# Intercellular adhesion boots collective cell migration through elevated membrane tension

Brent M. Bijonowski [1,4], Jongkwon Park [1,4], Martin Bergert[2], Christina Teubert[3], Alba Diz-Muñoz[2], Milos Galic [3] ✉ & Seraphine V. Wegner [1] ✉

In multicellular systems, the migration pattern of individual cells critically relies on the interactions with neighboring cells. Depending on the strength of these interactions, cells either move as a collective, as observed during morphogenesis and wound healing, or migrate individually, as it is the case for immune cells and fibroblasts. Mediators of cell-cell adhesions, such as cadherins coordinate collective dynamics by linking the cytoskeleton of neighboring cells. However, whether intercellular binding alone triggers signals that originate from within the plasma membrane itself, remains unclear. Here, we address this question through artificial photoswitchable cell-cell adhesions that selectively connect adjacent plasma membranes without linking directly to cytoskeletal elements. We find that these intercellular adhesions are sufficient to achieve collective cell migration. Linking adjacent cells increases membrane tension, which activates the enzyme phospholipase D2. The resulting increase in phosphatidic acid, in turn, stimulates the mammalian target of rapamycin, a known actuator of collective cell migration. Collectively, these findings introduce a membrane-based signaling axis as promotor of collective cell dynamics, which is independent of the direct coupling of cell-cell adhesions to the cytoskeleton.

Collective cell migration is a hallmark of morphogenesis, tissue homeostasis, and wound healing[1–3]. A central element that determines whether cells migrate collectively or individually is the strength of intercellular adhesions[4]. Cell-cell adhesions mediated by classical cadherins link neighboring cells through interactions between their extracellular domains, while α-catenin and β-catenin recruit to the cytoplasmic tail, forming a mechanosensitive link to the actin cytoskeleton[5,6]. Hence, cadherins both mechanically connect cells as structural elements and activate intercellular signaling circuits that regulate cytoskeletal dynamics and gene expression[7–12].

In contrast to signaling circuits emanating from protein complexes that link the cytoskeletons of adjacent cells, the contribution of signaling cues originating directly from the plasma membrane in collective cell dynamics has been largely overlooked. The membrane is often regarded as a passive structure whose primary role is to separate the cell's exterior from its interior, thereby enabling signal transduction via transmembrane protein complexes. However, recent studies challenge this view. In single cells, changes in membrane properties have been shown to significantly influence leading edge dynamics[13–15] and migration behavior[16–18]. Notably, these alterations in membrane properties are accomplished not only by direct mechano-chemical feedback loops, but also by intracellular signaling and changes in gene expression[19,20]. Consequently, changes at the plasma membrane not only yield short-term changes in cell, but also have long-term effects[21,22]. Collectively, these studies suggest that protein-independent signaling cascades that originate from within the

[1]Institute of Physiological Chemistry and Pathobiochemistry, University of Münster, Münster, Germany. [2]Cell Biology and Biophysics Unit, European Molecular Biology Laboratory, Heidelberg, Germany. [3]Institute of Medical Physics and Biophysics, University of Münster, Münster, Germany. [4]These authors contributed equally: Brent M. Bijonowski, Jongkwon Park. ✉e-mail: galic@uni-muenster.de; wegnerse@uni-muenster.de

plasma membrane are suitable for initiating sustained changes in cell dynamics. Despite this, although it is evident that membrane mechanics regulate the dynamics of individual cells, the contribution of plasma membrane-derived signals to collective cell migration—independent of the direct link between adhesions and cytoskeleton and the associated biochemical signaling cascades—remains unclear. A significant challenge in addressing this question is the lack of tools capable of dissecting the mechanical cues at the plasma membrane from the mechano-transduction through cell-cell adhesion molecules to the cytoskeleton and biochemical signals. For example, the truncation of the cytoplasmic catenin-binding domains renders E-cadherin nonfunctional, as the membrane-anchored extracellular domain alone is insufficient to sustain intercellular cell-cell adhesions[23,24].

Here, we set out to explore the relevance of membrane-based signaling for collective cell dynamics. To selectively elicit signals that originate from within the plasma membrane, we used artificial photoswitchable cell-cell adhesions based on the cyanobacterial phytochrome 1 (Cph1), which induce cell-cell adhesions on red light illumination without directly linking to the actin cytoskeleton[25]. We found that an increase in cell-cell connections leads to coordinated and collective cell movement in cells that otherwise would migrate as single cells. Increased cell-cell adhesions did not slow cells down, as would be expected when augmenting friction[26], but made them faster. At the cellular level, the photoactivated increase in cell-cell adhesions resulted in an increase in membrane tension. Notably, it also activated the enzyme phospholipase D2 (PLD2), resulting in an increase in phosphatidic acid (PA) and activation of mammalian target of rapamycin (mTOR) signaling. These findings demonstrate that mechanical coupling of cells at their membranes via artificial adhesion proteins that do not link directly to the actin cytoskeleton induces increased membrane tension, which is sufficient to achieve collective cell migration.

## Results

### Cph1-PM-mediated transcellular adhesion alters collective cell migration

To explore the contribution of signals that originate at the plasma membrane in collective cell migration, we engineered artificial photoswitchable cell-cell adhesion molecules. Specifically, we used the extracellular domain of the Cph1 protein from *Synechocystis* sp., which forms homodimers under red light (660 nm) and reversibly dissociates into monomers under far-red light (720 nm). To anchor Cph1 in the plasma membrane, we fused only Cph1 to the transmembrane domain of the platelet-derived growth factor receptor (Fig. 1a). We named the membrane-anchored artificial and photoswitchable adhesion molecule "Cph1-PM." Unlike classical cadherins, Cph1-PM lacks an intracellular domain for adapter proteins to bind to, preventing direct linkage of these artificial adhesions to the actin cytoskeleton. In general, when Cph1-PM is expressed on cell surfaces, the adhesions between them will form under red light and reverse under far-red light, as previously characterized[25]. Cph1 forms antiparallel dimers, as observed in its crystal structure[27], and therefore, when expressed on cell surfaces, it forms interactions in trans rather than in cis.

As a starting point, we used MDA-MB-231 (MDA) cells, which lack type-1 cadherins, including E- and N-cadherin; display weak cell-cell adhesions; and migrate as single cells[28]. To explore whether these artificial cell-cell adhesions can induce collective cell migration, we transfected MDA cells with Cph1-PM and generated monoclonal stable cell lines with Cph1-PM expression from them (Cph1-PM-MDA). Successful expression of Cph1 on the cell surface was confirmed by immunostaining unpermeabilized cells for the Myc-tag and subsequently analyzing them with flow cytometry and confocal microscopy (Supplementary Fig. 1a, b). In addition, we used MCF-7 cells as a positive control for a cell type that displays high expression levels of type-1 cadherins, strong cell-cell adhesions, and collective cell

migration[29]. To probe for collective and single-cell migration, we used wound-healing assays and set a baseline for the migration behavior by using wild-type MDA and MCF-7 cells. We observed similar wound closure rates in MCF-7 and MDA cells (Fig. 1b). However, the correlation length was higher in MCF-7 cells (Fig. 1c), indicating a greater degree of coordinated cell movement in this cell line. We further observed individual MDA cells at the wound edge depart from the confluent cell sheet, indicative of the weak intercellular adhesions between them, while the leading edge of the MCF-7 wound remained intact (Supplementary Fig. 1c and Supplementary Movie 1).

Next, we probed the effect of increased intercellular adhesions on MDA cell migration. For this, we compared the cell migration behavior of Cph1-PM-MDA cells under red light, where the artificial cell-cell adhesions are active, to cells under far-red light, where the artificial cell-cell adhesions are inactive. Exposure to red light caused Cph1-PM-MDA cells to migrate collectively and with a unified front (Fig. 1d and Supplementary Movie 2). Velocity vector analysis of cell migration revealed that, under red light, a greater proportion of vectors were oriented perpendicular to the wound edge compared with those observed under far-red light (Supplementary Fig. 1d, e). On the other hand, the orientation of the velocity vectors of Cph1-PM-MDA cells under far-red light and wild-type MDA cells was random (Fig. 1d, Supplementary Fig. 1c). Moreover, we found a significantly higher correlation length for Cph1-PM-MDA cells under red light than under far-red light, which had a comparable correlation length to the parent MDA cells (Fig. 1c). Notably, the correlation length of Cph1-PM-MDA cells under red light and that of MCF-7 cells was comparable. Finally, to ensure that light illumination did not influence cell migration, we performed a wound-healing assay under various light conditions. We found that the wound-healing rate and correlation length remained unchanged under red and far-red light compared with those under darkness, suggesting that illumination itself had no effect on cell migration (Supplementary Fig. 1g, h).

Several studies indicate that the strength of intercellular connections significantly affects collective cell dynamics (e.g., glass transition[30,31], plithotaxis[32,33]), whereby "fluidification" resulting from reduced intercellular friction is generally associated with increased cell speed and a decrease in correlation length. Counterintuitively, the wound closed faster for Cph1-PM-MDA cells under red light than it did either under far-red light or in MDA cells lacking the construct (Fig. 1b), despite increased friction between the cells and the cells being more restricted in their freedom to move independently. These differences in wound closure rate were consistent over the entire 12-h duration of the experiment (Supplementary Fig. 1f). To further analyze the origin of the faster wound closure rate under red light, we performed single-cell motion analysis (Fig. 1e and Supplementary Movie 3). The traces of individual Cph1-PM-MDA cells under red light exhibited a directional migration toward the wound during the analyzed 16-h time window, whereas Cph1-PM-MDA cells under far-red light showed frequent random trajectories. For analysis, cells were grouped on the basis of localization into "front cells" (i.e., 3 cell diameters from the wound edge) and "rear cells" (Supplementary Fig. 2a). Under red light, Cph1-PM-MDA cells at the "front" moved more directly toward the wound compared with cells under far-red light (Fig. 1f, g, Supplementary Fig. 2b, c). Moreover, also at the single-cell level, Cph1-PM-MDA cells at the front migrated faster under red light than under far-red light (Fig. 1h, i). Finally, to account for the possibility that photoactivation by itself changed cell dynamics, we cultured Cph1-PM-MDA cells at low density and tracked individual cells on exposure to red and far-red light. We found no differences in cell shape, cell division, or cell speed (Supplementary Fig. 3a–f). Collectively, these experiments suggest that the observed changes in collective cell dynamics are due to changes in transcellular and not intracellular dynamics.

The photoregulation of the Cph1-based cell-cell adhesions enable dynamic and reversible control over these adhesions and the resulting

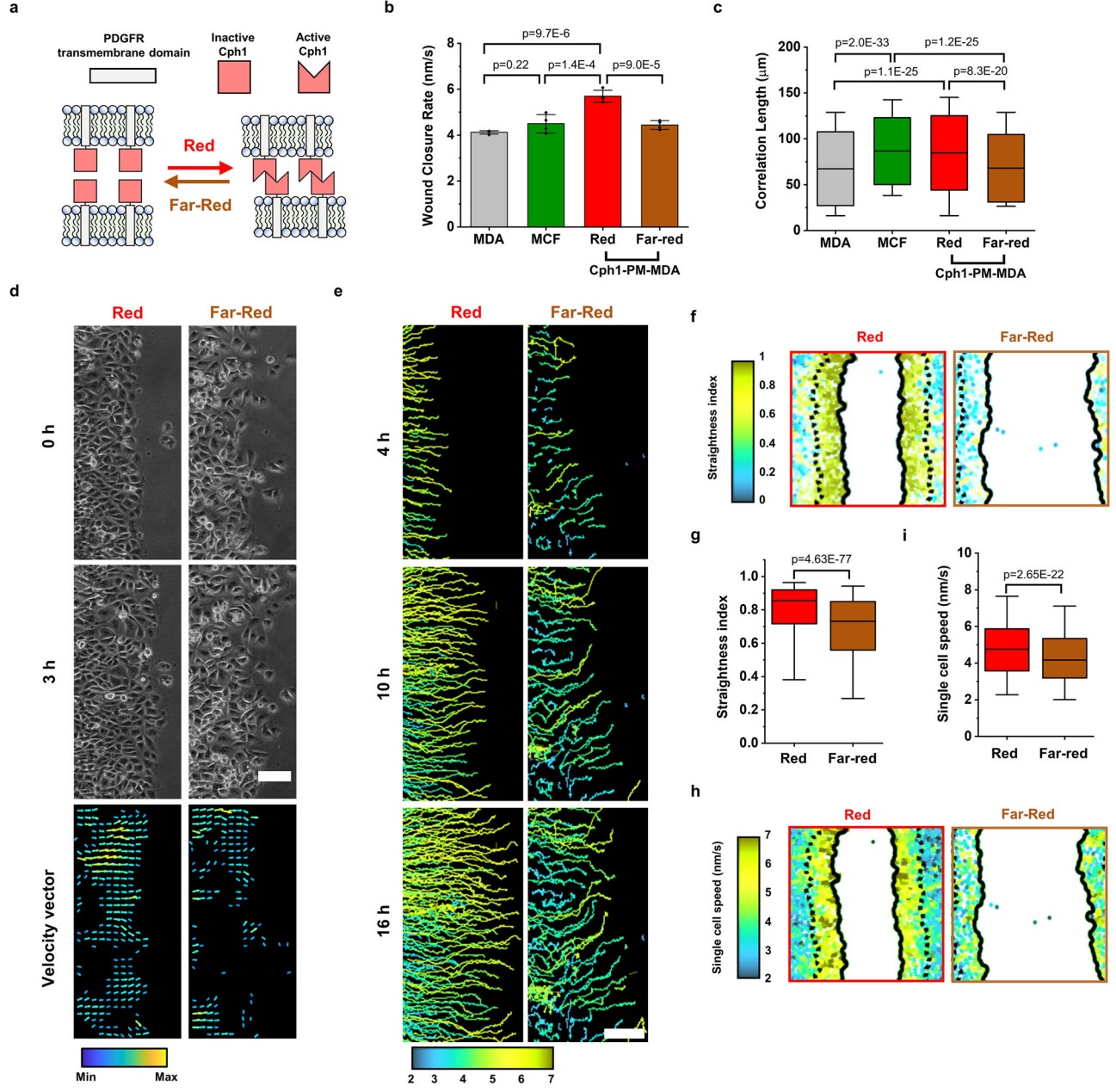

**Fig. 1 | Cph1-PM activation alters cellular migration and coordination.**
**a** Schematic of Cph1-PM-based adhesion. **b** Average wound closure rate of parental MDA ($n = 4$), MCF7 ($n = 4$), and Cph1-PM-MDA under red ($n = 4$) and far-red ($n = 4$) light from 4 biological replicates (one-way ANOVA with Tukey's multiple comparisons test). **c** Correlation length denotes the persistence length of the velocity vector components of MDA ($n = 3443$), MCF7 ($n = 793$), and Cph1-PM-MDA under red ($n = 732$) and far-red ($n = 1553$) light from 4 biological replicates (one-way ANOVA with Scheffe's multiple comparisons test). **d** Phase-contrast micrographs (top) and velocity vector maps (bottom) depicting cell migratory trajectory of Cph1-PM-MDA cells under red (left) and far-red (right) light. On the vector map, yellow indicated the maximum and blue the minimum. **e** Single-cell tracking by using nuclei to evaluate cell movement trajectory. Cell trace showing movement for first 4, 10, and 16 h. Blue represented 2 nm/s, yellow indicates 7 nm/s, with other colors reflecting intermediate speed. **f** Plots of Cph1-PM-MDA cell tracks with color

code for straightness index under red or far-red light. Blue represented 0, yellow indicates 1, with other colors displaying intermediate index. **g** Straightness index, which evaluates the efficiency of wound healing, from 2 to 4 h on Cph1-PM-MDA under red ($n = 2823$) or far-red ($n = 2122$) light from 2 biological replicates (two-tailed, two-sample Kolmogorov-Smirnov test). **h** Plots of Cph1-PM-MDA cell tracks with color code for single-cell speed under red or far-red light. Blue represented 2 nm/s, yellow indicates 7 nm/s, with other colors reflecting intermediate speed. **i** Single-cell speed based on cell tracking from 2 to 4 h under red ($n = 2377$) and far-red ($n = 1956$) light from 2 biological replicates (two-tailed, two-sample Kolmogorov-Smirnov test). Bar plots (**b**) are denoted as the mean with standard deviation. Box plots (**c**, **g**, **i**) present the median and 25th and 75th percentiles, and lower and upper boundaries of whiskers show the 5th and 95th percentiles. Scale bars (**d**, **e**), 100 μm. Data source and statistical details are provided as a source data file.

cell migration. To investigate this, we examined whether switching illumination from red light to far-red light, or vice versa, would alter the migration behavior. In the wound-healing assay, we illuminated Cph1-PM-MDA cells with red light for the first 6 h, followed by far-red

light for the next 12 h (Supplementary Fig. 4a, Supplementary Movie 4). We observed that the wound-healing rate began to decrease immediately after switching from red to far-red light. Moreover, the average wound-healing rate and correlation length both decreased following

the change in illumination (Supplementary Fig. 4b, c). On the other hand, when the cells were exposed to far-red light for 6 h followed by red light for 12 h, the wound-healing rate did not significantly change after switching the illumination (Supplementary Fig. 4d, e, Supplementary Movie 5). At the same time, the correlation length did not increase under red light illumination until the end of the experiment (Supplementary Fig. 4f), indicating a lack of collective and coordinated movement. We reason that the cells had already dispersed during the initial 6 h of far-red light exposure. Although Cph1 at the molecular level is activated within milliseconds of red light illumination[34], cells must be in close proximity for cell-cell adhesions to form and for collective cell migration to emerge. Likewise, Cph1 is also deactivated within milliseconds of far-red light activation, but it takes longer for cells to separate from each other.

## Cph1-PM-mediated transcellular adhesion augments plasma membrane tension

Our data show that increased intercellular adhesion in Cph1-PM-MDA cells augments cell speed and directionality at the same time (Fig. 1b, h). Since Cph1-PM lacks a cytosolic tail that could directly interact with intracellular proteins, we hypothesized that Cph1-PM may trigger signaling that originates from within the plasma membrane. To test this assumption, we probed for changes in membrane properties.

First, we measured whether Cph1-PM activation influenced lipid ordering. For that, we took advantage of the hydrophobic fluorescence membrane dye Pro12A, which exhibits a blue-shifted emission with increased lipid ordering. In this ratiometric imaging, a rise in generalized polarization (GP) is associated with stiffer gel-like lipid membrane structure and a decrease is associated with increased fluidity[35]. Focusing on contact areas between two cells in a confluent layer, we observed an increase in GP when Cph1-PM was activated with red light for 15 min (Fig. 2a). In contrast, when the cells were first exposed to red light for 15 min and then to far-red light for 15 min, to inactivate Cph1-PM-based adhesions, the GP values decreased (Fig. 2b). The changes in GP were fast and first responses were observed within 5–10 min at the time resolution of the imaging (Supplementary Fig. 5a). Moreover, similar trends in GP values were also observed for cells at the wound edges when switching from red to far-red light and vice versa (Supplementary Fig. 5b). As activation of Cph1-PM-based cell-cell adhesions can be modulated by varying red light intensities (Supplementary Fig. 6a, b), we consistently used an intensity of 1440 $\mu$W/cm$^2$ throughout the whole study, which is significantly higher than the 15 $\mu$W/cm$^2$ required for full activation.

The results to this point indicate that the direct physical cell-cell membrane adhesions locally alter membrane structure and make the interphase membranes stiffer at the cell-cell contact sites. To further analyze the alternations in membrane properties on induction of the artificial cell-cell adhesions, we next performed static tether pulling by using atomic force microscopy (AFM) to examine changes in apparent plasma membrane tension at the cellular scale. Specifically, a membrane tether on cells positioned close to the wound edge was maintained at a constant distance until it ruptured, and the corresponding force difference was quantified (Fig. 2c). We found that Cph1-PM-MDA cells had a significantly higher static tether force under red light than did these cells under far-red light (Fig. 2d). Moreover, we did not observe significant differences in membrane tension in the front versus the rear of the wound for either condition (Supplementary Fig. 5c, d), which argues against profound tension gradients between different cells.

## Cph1-PM-mediated changes in membrane tension, membrane-to-cortex attachment, and formation of focal adhesions

The static tether force measured in the previous experiments is proportional to the apparent membrane tension, which integrates the in-plane tension of the lipid bilayer, as well as the membrane-to-cortex

attachment[17,36,37]. To investigate the origin of the observed difference in apparent membrane tension, we next probed the membrane-cortex interface. Previous studies showed that phosphorylation of Ezrin/Radixin/Moesin proteins (pERM) strengthens the connection between F-actin and the plasma membrane, thereby increasing the apparent membrane tension (Fig. 2e)[38]. To probe for changes in pERM on Cph1-PM activation, we exposed Cph1-PM-MDA cells to either red or far-red light, fixed the cells under the respective illumination, and then stained for F-actin, pERM, and DAPI (Fig. 2f). Here, we found a quantitative increase of the pERM signal in Cph1-PM-MDA cells under red compared with far-red light (Fig. 2g) as another indicator of increased membrane tension on the formation of artificial cell-cell adhesions. At the same time, we observed an accumulation of actin stress fibers along the cell edge under red light, as well as an increase in phosphorylated myosin, particularly in cells at the wound edge, suggesting enhanced contractility (Supplementary Fig. 5e)[39].

Considering that membrane tension has been reported to affect focal adhesion in single cells[40], we investigated how cell membrane adhesion affected cell spreading and the number of focal adhesions[41,42]. Staining the focal adhesion-associated protein vinculin showed an increase in the number of focal adhesions in Cph1-PM-MDA cells under red light compared with those under far-red light (Fig. 2h, l). Although the total area of the focal adhesions per cell increased, the cell spreading area and the area of individual focal adhesions between Cph1-PM-MDA cells under red and far-red light did not show significant differences (Fig. 2i, j, k). Previous reports showing that not the total number, but the size, of the focal adhesions is the determining factor in cell speed suggest that the increased number of cell adhesions under red light compared with far-red light would not slow down cell migration[43].

## Cph1-PM-mediated effects are independent of E-cadherin signaling

Having observed collective migration, a feature generally associated with epithelial cell types, in cells originating from a mesenchymal cell type (MDA), we investigated changes in the expression of epithelial and mesenchymal markers. We found no expression of the cell-cell adhesion molecule E-cadherin, a marker of the epithelial phenotype, in Cph1-PM-MDA cells independent of light treatment, whereas it was clearly present in control MCF7 cells (Fig. 3a).

The observed changes in Cph1-PM-MDA cell dynamics on exposure to red light raises the question of what downstream signaling pathways are triggered. First, to determine the role of actin polymerization in Cph1-PM-dependent signal propagation, we added cytochalasin D (CytoD, 100 nM), an inhibitor of actin polymerization (Fig. 3b), and jasplakinolide (Jasp, 25 nM), an actin nucleation promoter (Fig. 3c), to the wound-healing assays. Because of the strong cell toxicity of CytoD, its concentration was optimized to remain below toxicity while still preventing actin polymerization (Supplementary Fig. 7a–c). Treatment with CytoD reduced the wound closure rate in Cph1-PM-MDA cells both under red and far-red light, confirming the central role of actin in cell migration (Fig. 3b, d). Addition of Jasp under far-red light yielded no significant changes compared with untreated cells, but increased the wound-healing rate under red light (Fig. 3c, d). At the same time, the correlation length increased in Jasp-treated cells under both red and far-red light, but the difference between the two illumination conditions remained stable (Fig. 3e).

Next, we evaluated Cph1-PM-dependent changes in the mechanosignaling associated with E-cadherin. E-cadherin has been described to activate YES associated protein (YAP)-dependent gene expression with changes in mechanical strain[44]. When evaluating the localization of YAP under different illumination, we found no difference in the cytoplasmic versus nuclear localization (Fig. 3f, g). Consistently, expressions of two YAP down-streaming markers, TEA domain transcription factor 1 (TEAD1) and connective tissue growth factor (CTGF),

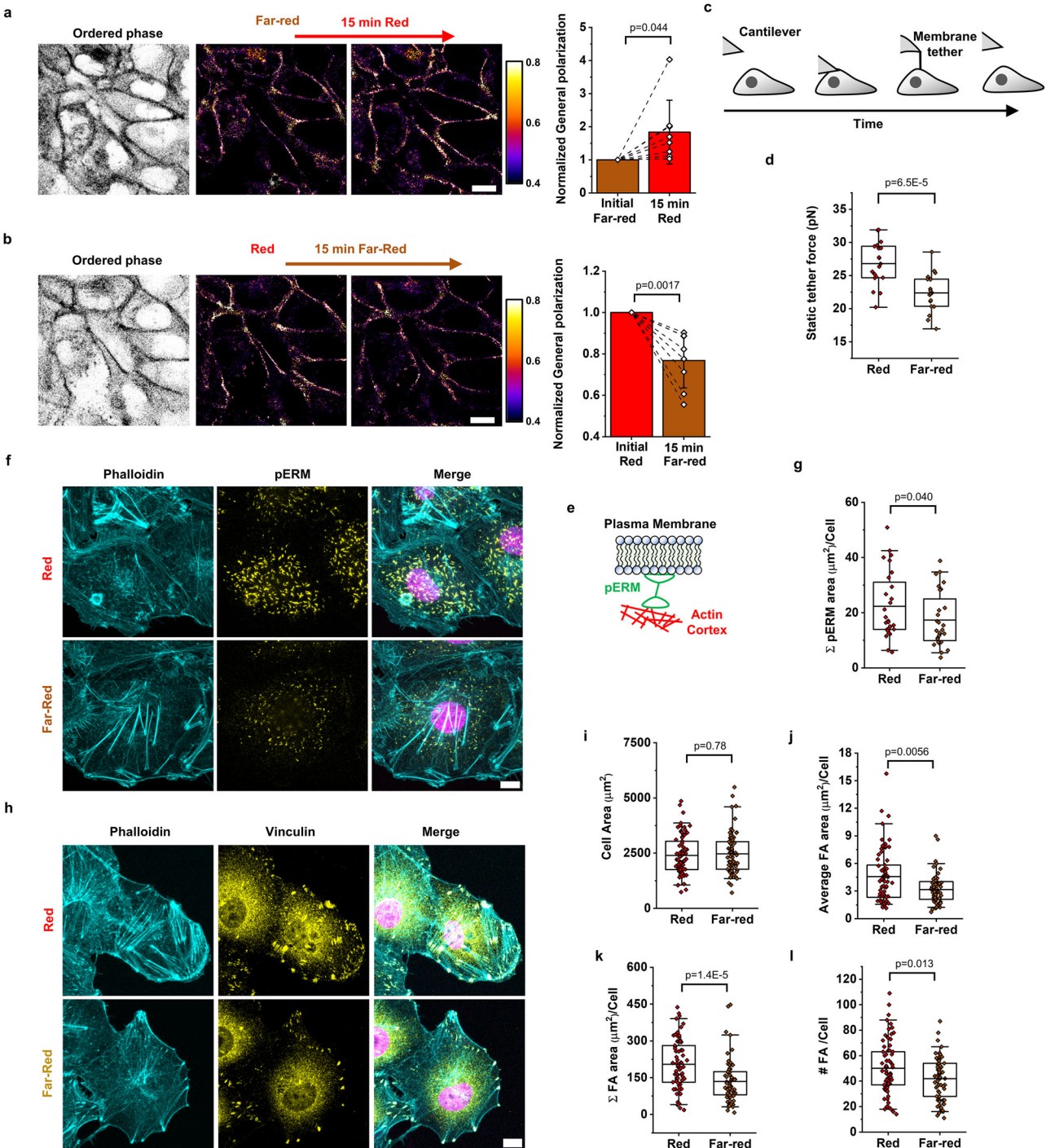

**Fig. 2 | Activation of Cph1-PM augments membrane tension, ERM, and focal adhesion expression.** Pro12A staining shows polarization changes on changing illumination (**a**) from far-red to red light and (**b**) from red to far-red light. Ordered phase images were used to detect the cell boundaries. Plots depicting the normalized general polarization at the cell-cell interface on changes from far-red to red light (a; *n* = 8) or from red to far-red light (b; *n* = 8) (*n* = 3 biological replicates; two-tailed, paired sample *t*-test). **c** Schematic showing how the static tether pulling force is measured by atomic force microscopy (AFM). **d** Static tether force of Cph1-PM-MDA cells exposed to far-red (*n* = 20) or red (*n* = 20) light (3 biological replicates, two-tailed, two-sample *t*-test). **e** Schematic representation of phosphorylation of Ezrin/Radixin/Moesin proteins (pERM) proteins. **f** Confocal fluorescence images of Cph1-PM-MDA cells stained with phalloidin (cyan), anti-pERM antibody (yellow), and DAPI (violet). **g** Quantification of pERM area per cell under far-red

(*n* = 32) and red (*n* = 32) light exposure (3 biological replicates, two-tailed, two-sample Mann-Whitney test). **h** Confocal fluorescence images of Cph1-PM-MDA cells stained with phalloidin (cyan), anti-vinculin antibody (yellow), and DAPI (violet). **i** Cell spreading area (two-tailed, two-sample Mann-Whitney test), **j** focal adhesion (FA) area per cell (two-tailed, two-sample Mann-Whitney test), **k** total FA area per cell (two-tailed, two-sample Mann-Whitney test), and **l** number of FAs per cell (two-tailed, two-sample *t*-test) of Cph1-PM-MDA cells exposed to either red (*n* = 69) or far-red (*n* = 67) light from 3 biological replicates. Bar plots (**a**, **b**) are reported as the mean and whiskers represent the standard deviation. The box plot (**d**, **g**, **i**, **j**, **k**, **l**) presents the median and the 25th and 75th percentiles. The lower and upper boundaries of the whiskers show the 5th and 95th percentiles. Scale bars (**a**, **b**, **f**, **h**) are 10 µm. Data source and statistical details are provided as a source data file.

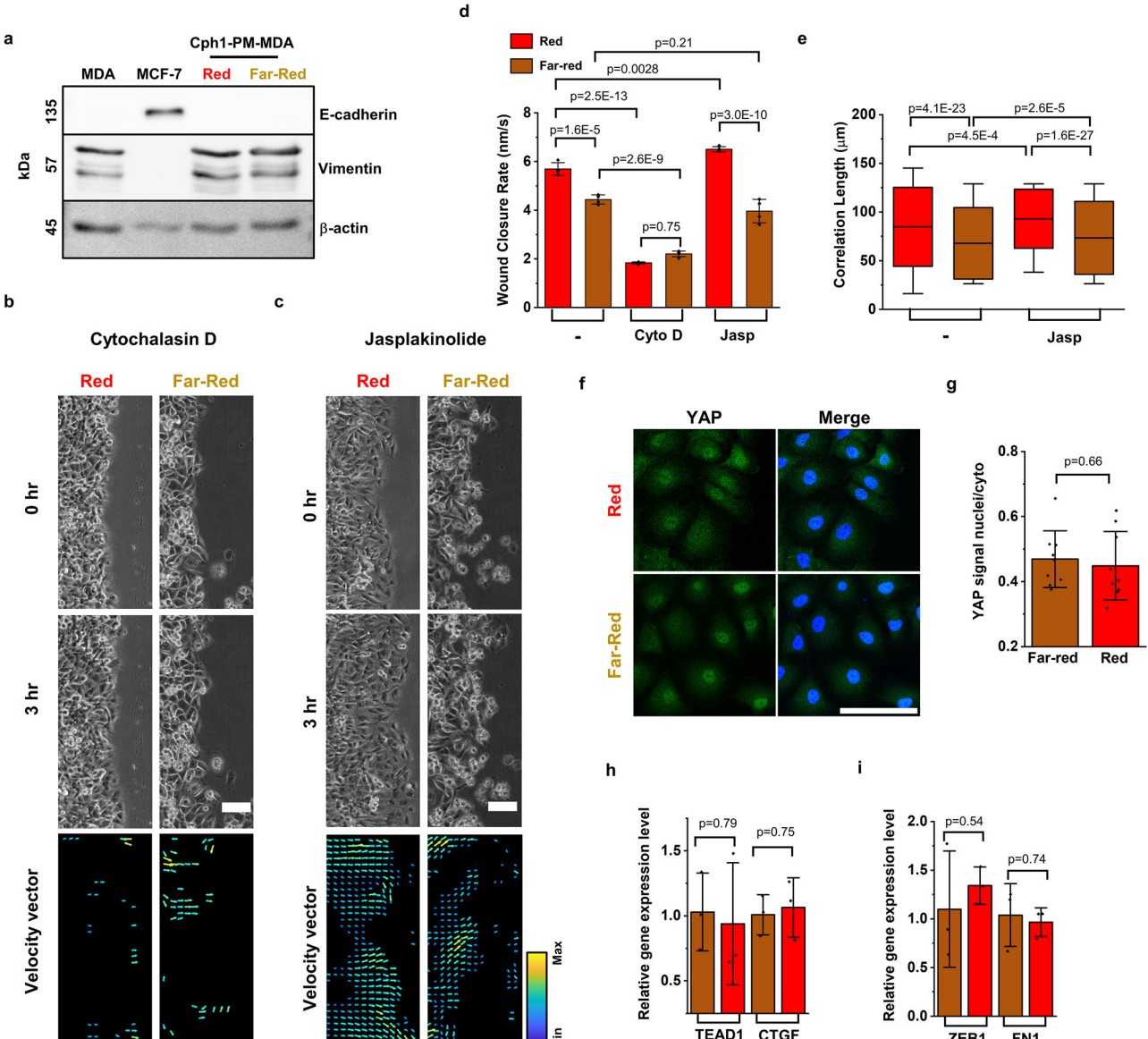

**Fig. 3 | Cph1-PM-dependent changes in collective cell dynamics do not change cell identity. a** Western blot analysis of E-cadherin and vimentin in parental MDA cells, MCF cells, and Cph1-PM-MDA cells exposed to red and far-red light. **b, c** Phase-contrast micrographs (top) and velocity vector maps (bottom) for Cph1-PM-MDA cells under red and far-red light on addition of **b** cytochalasin D (CytoD) and **c** jasplakinolide (Jasp). On the vector map, yellow indicated the maximum and blue the minimum. **d** Average wound closure rate of Cph1-PM-MDA without treatment under red ($n = 4$) and far-red ($n = 4$) light, with Cyto D under red ($n = 4$) and far-red ($n = 4$) light, or with Jasp under red ($n = 4$) and far-red ($n = 4$) light (one-way ANOVA with Bonferroni's multiple comparisons test). **e** Correlation length denotes the persistence length of the velocity vector of Cph1-PM-MDA under red ($n = 732$) or far-red ($n = 1553$) light, or with Jasp under red ($n = 533$) or far-red ($n = 2440$) light ($n = 4$ biological replicates; one-way ANOVA with Bonferroni's multiple comparisons test). **f** Immunofluorescent images of Cph1-PM-MDA cells under red (top) and

far-red (bottom) light stained against YES associated protein (YAP; green) and DAPI (blue). **g** Quantification of YAP localization in Cph1-PM-MDA cells under far-red ($n = 9$) and red ($n = 9$) light. Plot depicts nuclear vs cytoplasm intensity (3 biological replicates, two-tailed, two-sample $t$-test). **h** Gene expression of TEA domain transcription factor 1 (TEAD1) and connective tissue growth factor (CTGF) in Cph1-PM-MDA cells exposed to far-red ($n = 3$) and red ($n = 3$) light ($n = 3$ biological replicates, two-tailed, two-sample $t$-test). **i** Gene expression levels of zinc finger E-box binding homeobox transcription factor 1 (ZEB1) and fibronectin 1 (FN1) in Cph1-PM-MDA exposed to far-red ($n = 3$) and red ($n = 3$) light (two-tailed, two-sample $t$-test). Bar plots (**d, g, h, i**) are denoted as mean with standard deviation. Box plots (**e**) present the median and 25th and 75th percentiles, and the lower and upper boundaries of whiskers show the 5th and 95th percentiles. Scale bars (**b, c**), 100 µm; **f** 20 µm. Data source and statistic details are provided as a source data file.

showed no significant differences between red and far-red light (Fig. 3h).

Finally, we evaluated epithelial–mesenchymal transition markers that would be expected to be downregulated with increasing epithelial characteristics. Vimentin, a mesenchymal cell marker, showed no change in Cph1-PM-MDA compared with the parent cell type in the western blot (Fig. 3a). Likewise, the gene expression levels of two representative mesenchymal markers, zinc finger E-box binding

homeobox transcription factor 1 (ZEB1) and fibronectin 1 (FN1), were comparable with red and far-red light exposure, as observed with RT-PCR (Fig. 3i). Collectively, we did not find any evidence that increased Cph1-PM-based cell-cell adhesions with red light exposure alters the expression of genes that are associated with the transition from a mesenchymal to an epithelial cell type. Hence, these findings suggest that Cph1-PM signaling and the observed collectivity in wound healing did not rely on any cadherin-dependent circuits.

## Cph1-PM-mediated transcellular adhesion promotes PLD2 signaling

The findings to this point establish that activation of the artificial cell-cell adhesions results in substantial changes in membrane tension, as well as in faster and collective cell migration. At the same time, we found no changes in signaling cascades originating from classical E-cadherin signaling. Unlike previously used E-cadherin constructs with a truncated cytosolic tail, which cannot mediate cell-cell adhesions because of the missing interactions with the cytoskeleton, Cph1-PM is able to maintain stable cell-cell adhesions under red light without connecting to the cytoskeleton. Therefore, we reasoned that the origin of the signal resulting in the altered phenotype may reside in the plasma membrane. Indeed, membrane tension has been implicated in promoting biochemical signals at the single-cell level[18]. More precisely, increased membrane tension was reported to augment PLD2 activity, which breaks down phosphatidylcholine (PC) into PA and choline[18]. Downstream, PA modulates the properties of cells through activation of various signaling pathways, chief among them the mTOR pathway[45]. Considering its mechanosensitive properties, we thus probed for changes in PLD2 activity through the quantification of its product PA[19]. We observed, consistent with increased PLD activity, significantly elevated PA levels in Cph1-PM-MDA cells under red light compared with those in cells under far-red light and in the MDA cells lacking Cph1-PM (Fig. 4a). This increase in PA in Cph1-PM-MDA cells under red light was clearly due to increased cell-cell adhesions and not the consequence of Cph1-PM expression in MDA cells, as no significant differences in PA content was observed between Cph1-PM-MDA cells under far-red light and the parent MDA cells.

To further consolidate these findings, we modulated the PLD2 pathway by adding either propranolol (Pro, $1\,\mu M$), which prevents the degradation of PA through phosphatidic acid phosphatase (PAP)[46], or by adding VU-0285655-1 (VU, $5\,\mu M$), which inhibits PLD2 directly (Fig. 4b)[47]. Following treatment with Pro, PA levels in Cph1-PM-MDA cells under red light remained as high as those in untreated cells (Fig. 4a). In contrast, the addition of VU decreased PA levels in Cph1-PM-MDA cells exposed to red light to the level of cells under far-red light (Supplementary Movie 6). Other than this, no significant changes in PA levels were observed when Cph1-PM-MDA cells were treated with either Pro or VU under far-red light or in parent MDA cells. These findings are relevant, as they suggest that no additional PA is produced in Cph1-PM-MDA cells under red light illumination without the activation of PLD2 (Fig. 4a).

To connect the elevation in PA levels with the increased migratory capacity, as has been demonstrated for single cells[18,37,48], we evaluated the effect of altered PLD2 activity on wound closure rates and coordination of the cellular movement (Fig. 4c, d). In the presence of Pro, Cph1-PM-MDA cells under red light showed similar wound closure rates to those of Cph1-PM-MDA cells under far-red light and untreated MDA control cells. In contrast, the treatment with VU resulted in a significant decrease in wound closure rate under red light, whereas it did not alter the wound closure rate under far-red light (Fig. 4e). These results directly indicate that PLD2 activity and PA levels contribute to the improved wound closure rate and faster movement of individual cells observed for Cph1-PM-MDA cells under red light. This relationship is further supported by the decreased correlation length with VU treatment under red light, which is comparable to the correlation length of Cph1-PM-MDA cells under far-red light (Fig. 4f). Collectively, these results suggest not only that the PLD2 activation modulates the migration potential of MDA cells, but that Cph1-PM activates the PLD2 pathway.

Finally, we directly altered membrane tension in cells through the addition of oleic acid (OA, $50\,\mu M$), which inserts itself into the plasma membrane, resulting in a uniform elevation of membrane tension (Fig. 4g). The exogenous increase of membrane tension via OA led Cph1-PM-MDA cells under far-red light and parent MDA cells to

migrate at the same pace as Cph1-PM-MDA cells under red light (Fig. 4h, i). However, it should be noted that in the presence of OA, Cph1-PM-MDA cell migration was not as coordinated under far-red light as it was under red light due to the lack of cell-cell connections (Fig. 4j, Supplementary Movie 7). Previous studies have demonstrated that OA enhances the migration speed of individual MDA cells by elevating PLD2 levels and activating downstream mTOR signaling[49]. Consistent with these findings, our results show that artificial cell-cell adhesion influences cell migration by promoting both collective behavior and individual cell speed, driven by an increase in membrane tension.

## Cph1-PM-mediated membrane signaling affects the phosphoinositide 3-kinase/mTOR pathway

The results to this point suggest that the direct connection between cells through Cph1-PM increased membrane tension, activating PLD2, which in turn increased PA levels. To understand how this may affect cell dynamics, we investigated the role of PA-associated pathways (Fig. 5a). In particular, we considered the PA-dependent phosphoinositide 3-kinase (PI3K)/mTOR activity that was previously shown to affect cell migration[50]. To this end, we perturbed the PI3K/mTOR pathway with wortmannin ($0.2\,\mu M$), an inhibitor of PI3K[51], and rapamycin ($0.2\,\mu M$), an inhibitor of the mTOR complex[52]. After wortmannin treatment, the wound closure rate of Cph1-PM-MDA cells under red light significantly decreased and was comparable to that of cells under far-red light (Fig. 5b). Similarly, the addition of rapamycin led to a significant decrease in the wound closure rate, as well as detachment of individual Cph1-PM-MDA cells from the wound edge under red light, but had no effect under far-red light. At the same time, the correlation length decreased with both drugs, regardless of light illuminations (Fig. 5c). Complementarily, we evaluated single-cell tracks during the wound-healing assay under red and far-red light in conjunction with rapamycin treatment (Fig. 5d, Supplementary Fig. 8a, b). In the absence of any inhibitor, as already observed, the cell trajectories were straighter and the individual cells moved faster under red light than under far-red light. Notably, the wound closure rate decreased for Cph1-PM-MDA cells treated with rapamycin under red light, but not under far-red light. These data suggest that the observed increase in cell motility of the Cph1-PM-MDA cells, induced by the artificial cell-cell adhesions under red light, is mediated through the mTOR signaling pathway.

## Discussion

Intercellular coupling is crucial for the emergence of collective cell dynamics. It, therefore, does not surprise that several types of cell-cell adhesion proteins facilitate this intercellular cohesion. For example, cadherins at adherent junctions and claudins/occludins at tight junctions bridge the actin cytoskeleton of adjacent cells, while desmosomal proteins link intermediate filaments between cells[53]. In addition, although primarily known for cell-extracellular matrix (ECM) interactions, integrins were recently shown to also influence cell-cell adhesions[54]. Although distinct in structure, localization, and function, many of these proteins display not only adhesive properties, but also mechanosensitive cell signaling. For instance, stretching of alpha-catenin, which links adherent junctions to actin filaments, exposes cryptic binding sites for vinculin[55]. Similarly, talin, which participates in transferring ECM-based forces at focal adhesions to actin filaments, exposes—under load—a binding site for vinculin[56]. Intriguingly, recent studies showed that desmoplakin, which couples desmosomes to intermediate filaments, also undergoes conformational changes under mechanical stress[57]. The picture emerging from these and other studies is that differences in the expression level of these proteins define the distinct cell-specific adhesive and mechanosensitive signaling properties, which jointly determine the collective cell dynamics.

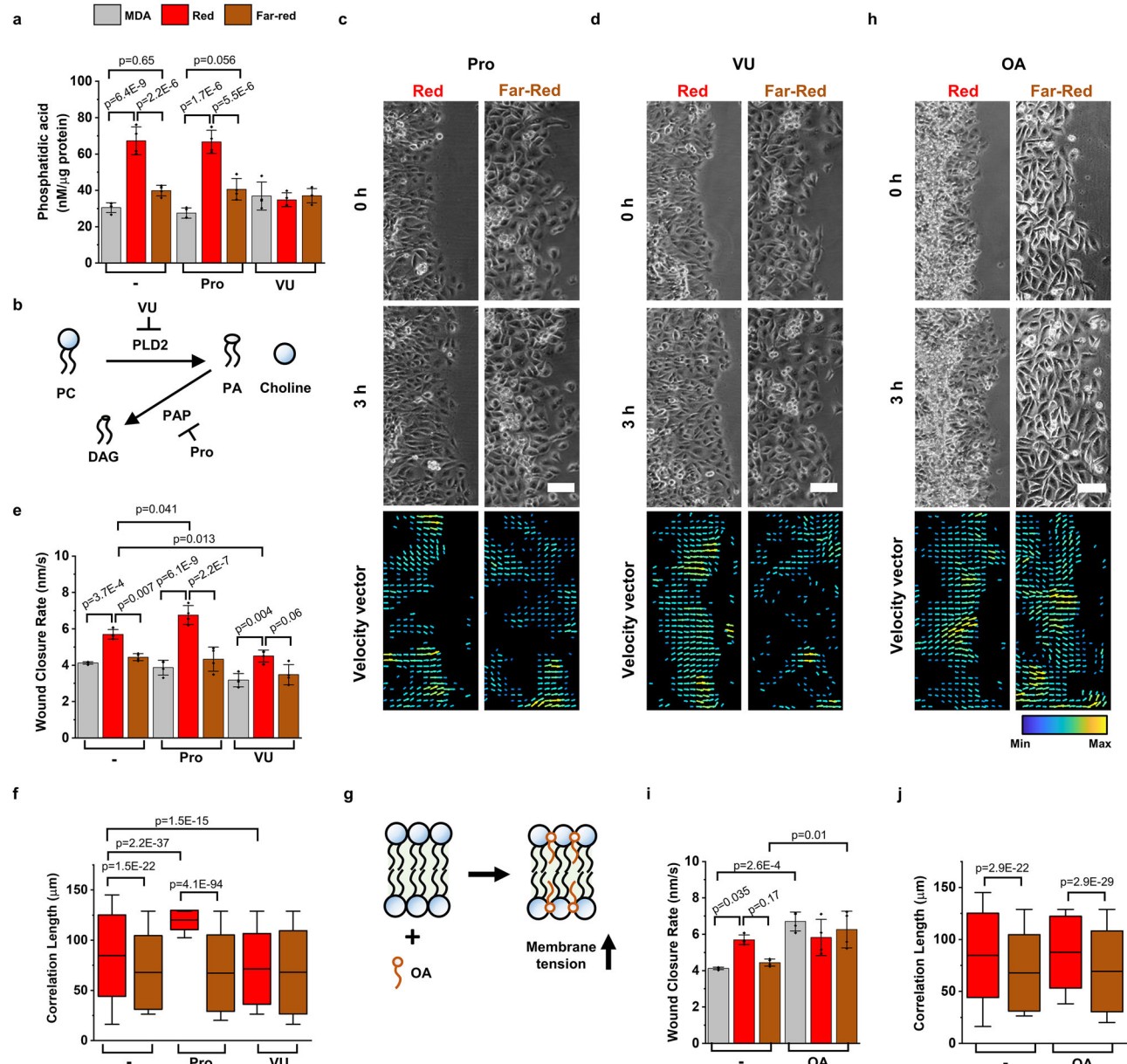

**Fig. 4 | Cph1-PM-dependent changes in collective cell dynamics are mediated by PLD2 activity and membrane tension. a** Phosphatidic acid levels for parental MDA and Cph1-PM-MDA cells under red and far-red light in the absence of or on addition of propranolol (Pro) or VU-0285655-1 (VU; all data $n = 4$ from 4 biological replicates). **b** Phospholipase D2 (PLD2) hydrolyzes phosphatidylcholine (PC) into phosphatidic acid (PA) and choline and phosphatidic acid phosphatase (PAP) hydrolyze PA to diacylglycerol (DAG). PAP is inhibited by Pro and PLD2 is inhibited by VU. **c**, **d** Representative phase-contrast micrographs (top) and velocity vector maps (bottom) for Cph1-PM-MDA cells under red and far-red light after treatment with Pro (**c**) and VU (**d**). On the vector map, yellow indicated the maximum and blue the minimum. **e** Wound closure rate from parental MDA or Cph1-PM-MDA cells under red and far-red light. For each condition, cells are shown before treatment or after treatment with Pro or VU (all data $n = 4$ from 4 biological replicates). **f** Correlation length of Cph1-PM-MDA cells without treatment under red ($n = 732$) and far-red ($n = 1553$) light, or after treatment with Pro under red ($n = 252$) and far-red ($n = 1588$) light, or after treatment with VU under red ($n = 2213$) and far-red

($n = 1209$) light from 4 biological replicates. **g** Oleic acid (OA) incorporates into the lipid bilayer of cell membranes and increases membrane tension. **h** Phase-contrast micrographs (top) and velocity vector maps (bottom) for Cph1-PM-MDA cells under red and far-red light after treatment with OA. On the vector map, yellow indicated the maximum and blue the minimum. **i** Wound closure rate from parental MDA cells or Cph1-PM-MDA under red and far-red light without treatment or after treatment with OA (all data $n = 4$ biological replicates). **j** Correlation length denotes the persistence length of velocity vector under red and far-red light before and after treatment with OA (no treatment, $n_{red} = 732$, $n_{far-red} = 1553$; OA, $n_{red} = 846$, $n_{far-red} = 1581$; 4 biological replicates). All statistics were analyzed by one-way ANOVA with Bonferroni's multiple comparisons test. Bar plots (**a**, **e**, **i**) are denoted as mean with standard deviation. Box plots (**f**, **j**) present the median and 25th and 75th percentiles, and the lower and upper boundaries of whiskers show the 5th and 95th percentiles. Scale bars (**c**, **d**, **h**), 100 μm. Data source and statistical details are provided as a source data file.

Complementing the current protein-centric view, in this study, we explored the role of the plasma membrane in this process. By using the artificial photoswitchable Cph1-PM-based cell-cell connections, we aimed to minimize the influence of other genetic or environmental

factors, ensuring that the observed differences in migration were primarily due to the connections between the cells. We found that augmented cell-cell adhesion results in collective cell migration at the multicellular scale and increased cell motility at the level of individual

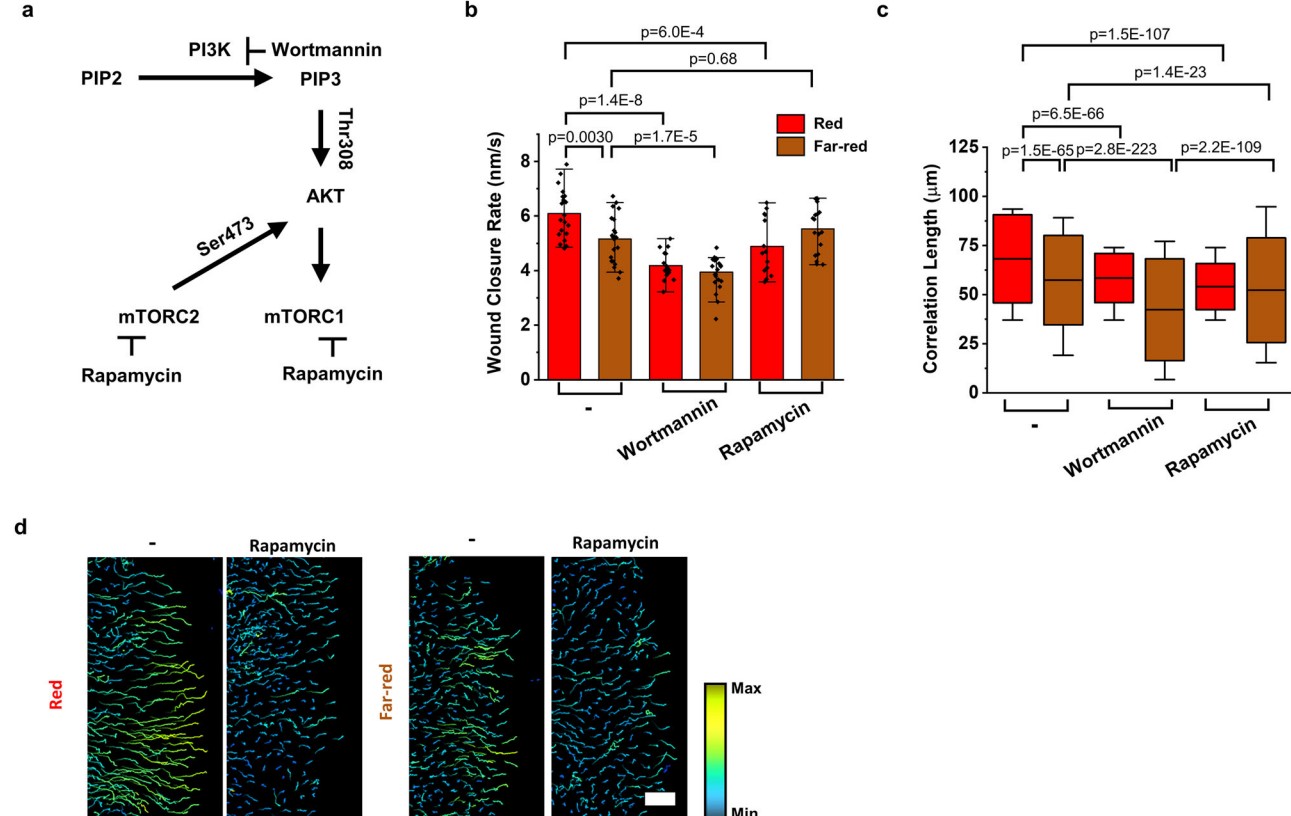

**Fig. 5 | Cph1-PM-dependent changes in collective cell dynamics are mediated through PI3K/mTOR signaling pathway. a** Schematics of phosphoinositide 3-kinase (PI3K) and mammalian target of rapamycin (mTOR) signaling pathway. PI3K is inhibited by wortmannin, and mTOR complex 1 (mTORC1) and mTOR complex 2 (mTORC2) are both inhibited by rapamycin. **b** Average wound closure rate of Cph1-PM-MDA without treatment under red ($n = 20$) and far-red ($n = 24$) light, with wortmannin under red ($n = 16$) and far-red ($n = 23$) light, or with rapamycin under red ($n = 14$) and far-red ($n = 10$) light. **c** Correlation length of Cph1-PM-MDA cells without treatment under red ($n = 1582$) and far-red ($n = 3237$) light, or after treatment with wortmannin under red ($n = 7608$) and far-red ($n = 5307$) light,

or after treatment with VU under red ($n = 3000$) and far-red ($n = 3542$) light from 3 biological replicates. **d** Single-cell tracking using nuclei to evaluate cell movement trajectory without treatment or with rapamycin under red or far-red light. Cell trace showing movement for 5 h. Yellow indicated the maximum and blue the minimum speed. Scale bars, 100 μm. Statistics were analyzed by one-way ANOVA with Tukey's (**b**) or Bonferroni's (**c**) multiple comparisons test. Bar plots (**b**) are denoted as mean with standard deviation. Box plots (**c**) present the median and 25th and 75th percentiles, and the lower and upper boundaries of whiskers show the 5th and 95th percentiles. Data source and statistical details are provided as a source data file.

cells. The increase in cell motility is associated with membrane ordering and a stiffer interface on formation of the artificial cell-cell adhesions. Notably, the resulting increase in membrane tension promotes further signaling. Specifically, we demonstrate that activation of PLD2 augments PA cleavage, which activates the mTOR/PI3K pathway to promote cell dynamics. These findings are consistent with experimental work obtained in single cells[18]. Hence, frictional forces at the plasma membranes initiate, analogous to mechanosensitive proteins under mechanical stress, a signaling cascade to synchronize collective cell dynamics (Fig. 6). However, although this study identifies constituents that mediate this collective cell motion, the precise sequence of signal propagation through the system remains elusive. Unlike the switch-like activation of Cph1-PM, which occurs at a timescale of milliseconds[34], the formation of cell-cell adhesions depends on the time that two cells require to come into contact. The resulting changes in membrane tension, which occur at timescales of minutes but may last for hours, will then affect systems that operate at faster timescales (e.g., steady-state turnover of the cell cortex and lipid metabolism), as well as those that operate at slower timescales (e.g., gene transcription and protein synthesis). Consequently, future work will be needed to delineate the exact cause and effect of the observed changes reported in this study.

Collectively, these findings establish that cell-cell connections alter membrane mechanics, which in turn alter collective migration. We consider this another regulatory layer that acts in parallel to protein-based systems to coordinate collective cell dynamics. Notably, although this study established a signaling cascade of Cph1-PM through tension-dependent activation of TOR signaling, it does not exclude the possibility that additional signaling cascades are activated. Given that Cph1-PM can be functionally replaced by any transmembrane protein, it is plausible to envision that the identified signaling circuit is of relevance for a broad spectrum of phenomena where mechanical stress is exerted on the plasma membrane.

## Methods
### Construct and sequence
The Cph1 domain of Cph1-PM was cloned into a pDisplay mammalian expression vector (Invitrogen V66020) between the Ig κ-chain leader sequence and the platelet-derived growth factor receptor domain as a Cph1-GFP fusion (pDisplay-GFP-Cph1)[25]. The N-terminal to the murine Ig κ-chain leader sequence directs the protein to the secretory pathway, while the C-terminal PDGFR transmembrane domain anchors the proteins on the extracellular side of the plasma membrane.

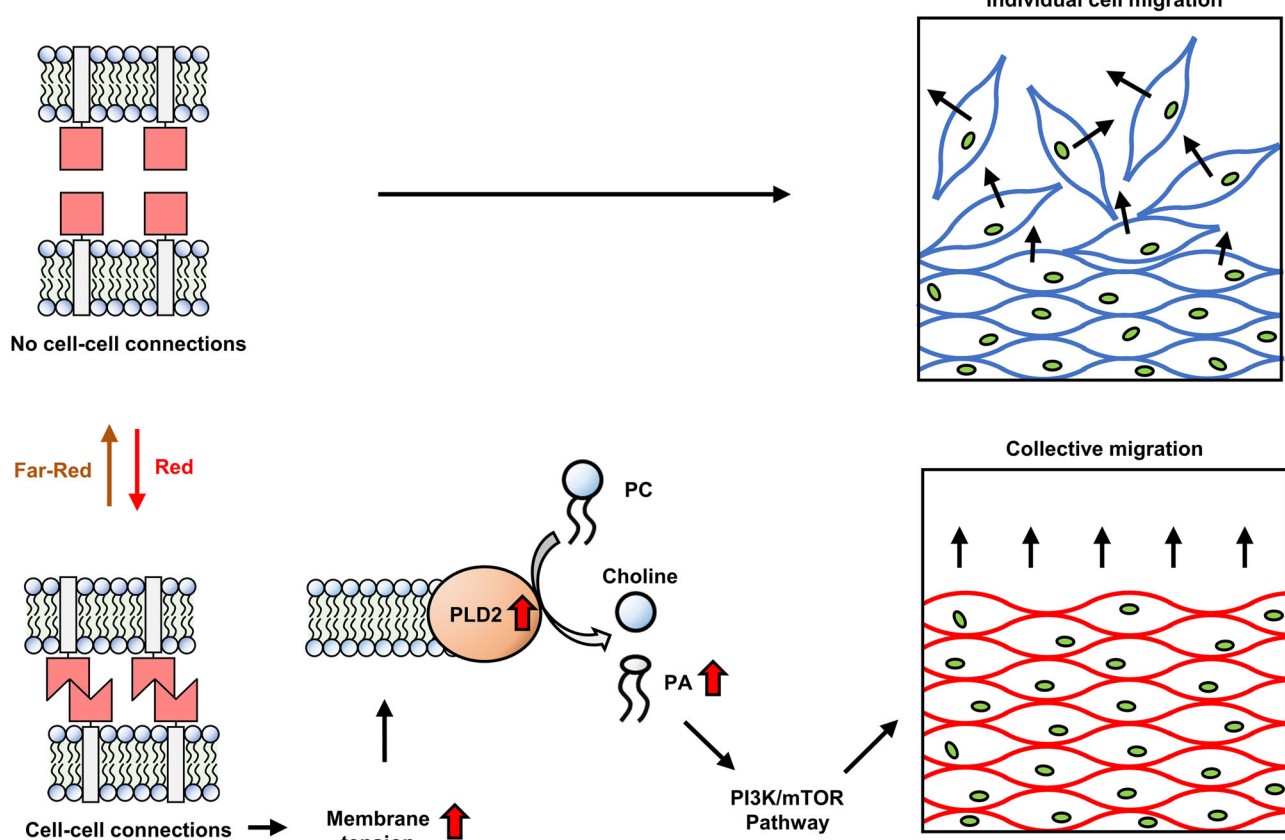

**Fig. 6 | Schematic of the proposed mechanism.** When cells are not connected, they migrate randomly. As cells become adherent, the physical adhesion increases cell membrane tension. The increase in membrane tension, in turn, promotes PLD2 activity, resulting in the degradation of PC into PA. The elevated PA activates the PI3K/mTOR pathway, promoting collective migration.

## Cell culture

MDA-MB-231 (HTB-26) and MCF-7 (HTB-22) cells were both purchased from ATCC (American Type Culture Collection). All cells were cultured in DMEM (Dulbecco's Modified Eagle Medium, Gibco) supplemented with 10% FBS (fetal bovine serum, Gibco), 1% penicillin/ streptomycin (Gibco), and 12.5 mM HEPES (Sigma Aldrich) at 37 °C and 5% $CO_2$. Cells were subcultured when they reached 80% confluence. A monoclonal stable cell line, Cph1-PM-MDA, that expresses Cph1-PM was generated starting from MDA-MB-231 cells[25]. The MDA-MB-231 cells were transfected with the pDisplay-GFP-Cph1 with Lipofectamine 3000 (Thermo Fisher Scientific, L300001), following the manufacturer's protocol. The transfected cells were selected with 1800 μg/mL G418 (Geneticin, Roche). After culturing the cells for 1 week, the signle cells were sorted by fluorescence activated cell sorting (BD FACS Aria cell sorter) into a 96-well plate with GFP signal. After several weeks, the monoclonal Cph1-PM-MDA were generated. Cph1-PM-MDA cells were maintained in 1800 μg/mL G418 (Geneticin, Roche) and used for up to 15 passages.

## Wound-healing assay

The wound-healing assay was performed in 24-well plates (Greiner Bio-One), into which MCF-7, MDA-MB-231, or Cph1-PM-MDA cells were seeded at $1.5 \times 10^5$ cells per well. Phycocyanobilin (PCB, 5 μM) was added to the Cph1-PM-MDA cells, and incubated overnight at 37 °C and 5% $CO_2$. Note that the activation of Cph1-PM is dependent on the presence of its cofactor PCB, as the protein remains inactive under red light in the absence of PCB treatment. 4 h before imaging, a vertical wound was created with a 200 μL tip, and fresh media containing the drugs (DMSO (control), Cytochalasin D (100 nM), Jasplakinolide

(25 nM), Propranolol (1 μM), VU-0285655-1 (5 μM), oleic acid (50 μM), wortmannin (200 nM), or rapamycin (200 nM) was added and kept in the media for the entire duration of the experiment. No additional PCB was added, since it would aggregate with the oleic acid and Cph1-PM domains were already saturated.

## Microscopy

An hour before imaging the multiwall plate was moved to an Ibidi multiplate heating system with gas control. The system was configured according to the manufacturer specifications (bottom glass 37 °C, cassette 38 °C, and top glass 42 °C), and maintained at 5% $CO_2$. Images were taken on an inverted Leica DMi8 microscope with Leica Application Suite X Ver. 3.7.1.21655. sotfware, using a 10× phase-contrast objective, every 10 min for 16 h. A 533/SP Bright Line HC short-pass filter (AHF Analysentechnik, 380–520 nm transmission) was used in front of the white light source to avoid photoactivation. Illumination was achieved with either an external red or far-red light lamp, which provided continuous illumination. The light intensities were 1440 μW/cm² with 620 nm for red light and 1120 μW/cm² with 734 nm for far-red light.

## Analysis of single and collective cell dynamics

For wound healing, images were analyzed with ImageJ (National Institute of Health, USA) to determine the area of the wound at each time point. The zero-time point was set at 4 h post wounding to avoid the effects resulting from variation in the cell seeding density and wounding process. Further, velocity vector analysis was carried out using bright field images in MATLAB version 7.10 (R2020a; Mathworks,

USA) and Particle image velocimetry (PIV) plugin from Dr. William Thielicke and Prof. Eize J. Stamhuis (MATLAB plugin). In our experiments, we applied PIV to track cell movement over time at regular intervals (10 min). Custom code was used to create correlation lengths (Supplementary Code 1), colored vector plots (Supplementary Code 2), and rose vector plots (Supplementary Code 3)[9].

To analyze the migration pattern during wound closure, single-cell tracking was performed using the TrackMate Plugin (v7.11.1)[58] in ImageJ (1.54 f). For particle detection and linking, a Laplacian of Gaussian (LoG) detector and a Simple Linear Assignment Problem Tracker were used, respectively. Detection of the wound edge and the analysis of migration parameters including cells speed, straightness index and the orientation of migration were performed in MATLAB (R2020b).

To distinguish between cells located close to the wound edge and cells further back in the field, a line parallel to the wound edge was drawn with a distance of 60 μm, which is approximately the size of three nuclei. If the center of a nuclei was located within this region, a cell was considered to be migrating at the front. Trajectories were cut off before cells from opposite sides of the wound were touching each other, trajectories shorter than four steps were excluded from analysis.

The straightness index was determined as the ratio between the net displacement and the total distance traveled, ranging from 0 (=movement back to the origin) to 1 (=straight trajectory). The cell speed represents the median speed of each cell in the respective time interval. As an indicator of directionality, the angle of migration in relation to the wound edge is used, referred to as the orientation angle. Angles around zero degree indicate migration towards the open space, the wound edge is here considered to be a straight vertical line.

For cell segmentation of single cells, unprocessed images were uploaded to Cellpose[59,60] with following parameters: model = cyto3; flow threshold = 0.75; cellprob threshold = 0.5; norm percentile lower = 10.0; norm percentile upper = 99.0; niter dynamics = 0; diameter = 50 pixels; image restoration = none. Upon segmentation, cell shape was determined in ImageJ using analyze particles. Speed was analyzed, as before, using the TrackMate Plugin.

For single-cell tracking analysis, Cph1-PM-MDA cells were stained with 0.5 μM Hoechst 34580 for 15 min and a wound was created with a 200 μL tip. Following, the cells were washed twice with PBS, and the media was replaced with DMEM containing 10% FBS with/without drugs. The Ibidi multiplate heating system and DMi8 microscope with the same settings as above (see "Microscope") were employed for image acquisition. The active form is of Cph1-PM is measured in the presence of PCB under red light, while the inactive form measured is in the presence of PCB under far-red light or without PCB under red light. Bright-field and nucleus images at a wavelength of 405 nm were captured every 10 min over a period of 16 h.

### Pro12A staining
Pro12A was generously gifted by Dr. Andrey S. Klymchenko, University of Strasbourg. Cph1-PM-MDA cells were seeded at $1 \times 10^4$ cells/cm² into a glass coverslip bottomed Ibidi m-dish 35 mm with PCB cofactor and incubated overnight to ensure adhesion. 45 min before imaging, 5 μM Pro12A was added and the plate was wrapped in aluminum foil to prevent illumination. Plates were then pre-incubated in either red or far-red light for 60 min before being imaged on an SP8 Leica microscope at 63× magnification. Fluorescent images were gathered by using 405 nm excitation with images in two bands (445 ± 15 nm and 525 ± 25 nm). Illumination was switched after 15 min to generate the transition from red to far-red and far-red to red light, resistively and maintained for another 15 min. Generalized polarization (ΔGP) was determined with ImageJ (Supplementary Code 4). The intensity at each pixel was compared using

the following equation and then normalized to the initial value:

$$\Delta GP = \frac{(I_{445} - GI_{525})}{(I_{445} + GI_{525})} \tag{1}$$

$$G = \frac{G_{ref} + G_{ref}G_m - G_m - 1}{G_m + G_{ref}G_m - G_{ref} - 1} \tag{2}$$

G is the correction factor, $G_{ref}$ is a reference value associated with the dye (0.207), and $G_m$ is the measured ΔGP for the dye dissolved in DMSO. The presented protocol was adapted from Owen et al.[61].

### Atomic force microscopy
Cph1-PM-MDA and MDA-MB-231 cells were seeded at $5 \times 10^5$ cells into a Fluorodish (WPI) and incubated overnight with 5 μM PCB and 1800 μg/mL G418 at 37 °C and 5% CO2. A wound was created with a 200 mL tip and washed twice with PBS. The cells were incubated with DMEM with 10% FBS for 1–2 h. 30 min before the AFM experiment, the media was replaced with DMEM containing 2% FBS. For the first 10 min and throughout the AFM measurements, Cph1-PM-MDA cells were continuously activated or inactivated using 635 nm or 740 nm light. qp-SCONT cantilevers (Nanosensors) were mounted on a CellHesion 200 AFM (Bruker), connected to an Eclipse Ti inverted light microscope (Nikon). Cantilevers were calibrated using the contact-based approach, followed by coating with 3 mg/ml Concanavalin A (Sigma) for 1 h at 37 °C. Cantilevers were washed once with PBS before the measurements. Apparent membrane tension was estimated using static tether pulling as follows: Approach velocity was set to 0.5 μm/s, with a contact force of 200 nN, and contact time was varied between 100 ms to 10 s, aiming at maximizing the probability of extruding single tethers. To ensure tether breakage at 0 velocity, the cantilever was then retracted for 10 μm at a velocity of 10 μm/s. Afterwards, the Z-position was kept constant for 25 s and tether force at the moment of tether breakage was recorded at a sampling rate of 2000 Hz. The resulting force-time curves were analyzed using JPK Data Processing Software. Measurements were run at 37 °C with 5% CO₂ and samples were used no longer than 2 h for data acquisition. More details can be found here: Bergert et al.[36].

### Phosphatidic acid assay
Total PA was measured with an assay kit purchased from Cell Biolabs, Inc. and run according to the manufacturer's instructions. In brief, $5 \times 10^6$ cells were plated in triplicate in the presence of PCB and either with or without PLD2 modulating drugs, and cultured overnight. The cells were illuminated for 4 h before the cells were washed three times with ice-cold PBS for 5 min each time. Then, cells were collected via scraping over ice. Cells were pelleted and resuspended in 1 mL of PBS before being sonicated for 30 s at 1 s pulses at 30% power. For lipid extraction, 1.5 mL methanol, 2.25 mL 1 M NaCl, and 2.5 mL chloroform were added, vortexed, and centrifuged at $1500 \times g$ for 10 min at 4 °C. The upper aqueous phase was discarded, and the lower chloroform phase was washed twice with pre-equilibrated upper phase, then transferred to a vial. The solvent was removed from the samples and then resuspended in 50 μL an assay buffer. 10 μL of each sample was transferred to a 96 well plate and incubated with 40 μL of Lipase Solution at 37 °C for 30 min. 50 μL of the Detection Enzyme Mixture provided with the kit was added and the samples were incubated at room temperature for 10 min in the dark. The concentration of PA was determined by measuring the fluorescence intensity using excitation at 545 ± 15 nm and emission at 590 ± 5 nm. The PA contents were calculated based on PA standards provided by manufacturer.

### Antibody staining
$5 \times 10^5$ cells were plated onto $22 \times 22$ mm glass coverslips placed in each well of a 6-well plated and incubated overnight with PCB at 37 °C

and 5% $CO_2$. A vertical wound was formed, using a 1 mL pipet tip, the coverslips were twice washed with PBS and incubated with growth media with G418 and PCB 5 μM under the proper illumination condition for 4 h. And then, they were fixed with 4% paraformaldehyde. Coverslips were then blocked with blocking and PERM buffer (0.2% BSA + 0.1% Triton X-100 in PBS) for 60 min at room temperature. Then, the coverslips were incubated with 1:800 rabbit-anti-YAP antibody (#14074, Cell signaling), 1:200 rabbit-anti-pERM antibody (#3726, Cell Signaling), 1:100 mouse-anti-Vinculin monoclonal antibody (13-9777-82, Invitrogen) in blocking and PERM buffer overnight at 4 °C. Coverslips were triple washed with blocking and PERM buffer and incubated with blocking and PERM buffer containing secondary antibody, 1:1000 Alexa 488 anti-rabbit (#4412, Cell Signaling) or 1:1000 Alexa 488 anti-mouse (#A11029, Invitrogen), 0.1 μg/ml of Phalloidin-TRITC (#ab176757, Abcam), and 1 μg/ml DAPI (#D1306, Invitrogen) for 2 h at room temperature. The secondary was removed, and the coverslips were triple-washed for 5 min each with PBS and mounted on glass slides for imaging.

To measure the area of vinculin and pERM staining in cells, the cell boundaries were defined using phalloidin staining. The vinculin and pERM signals were analyzed in each cell according to established protocols[62]. To improve image analysis, the background was first removed, and local contrast was enhanced using CLAHE (Contrast Limited Adaptive Histogram Equalization). An exponential function (exp) was then applied to further minimize background interference. Brightness and contrast were automatically adjusted to optimize feature visibility. Finally, the grayscale image was converted into a binary format, and objects with an area greater than 0.5 μm² were identified and counted for each cell.

For YAP analysis, a mask was created using DAPI staining. The intensity of the YAP inside and outside the nuclei was measured using the DAPI mask, and the nuclear signal was divided by the cytoplasmic signal.

### Antibody staining without permeablization
For unpermeabilized immunostaining, we used live, unfixed cells. First, $5 \times 10^5$ cells were plated on 6-well plate, and incubated at 37 °C and 5% $CO_2$. Then cells were blocked with blocking buffer for 5 min, and were incubated with 1:250 mouse-anti-myc-tag monoclonal antibody (#2276, Cell Signaling) in blocking buffer at room temperature. The cells were washed with blocking buffer twice, and incubated with 1:1000 Alexa 488 anti-mouse (#A11029, Invitrogen) and 1:2000 SYTO™ Deep Red Nucleic Acid Stain (S34900, Invitrogen) in blocking buffer for 1 h at room temperature. Finally, the cells were washed twice with blocking buffer and twice with PBS, the pictures were obtained with Leica SP4 confocal microscope.

For flow cytometry, $5 \times 10^5$ cells were transferred to 1.5 ml of centrifuge tube. The cells were incubated with 1:250 mouse-anti-myc-tag monoclonal antibody (#2276, Cell Signaling) in PBS for 1 h at 4 °C. The cells were washed twice with PBS by using centrifugation and incubated with 1:1000 Alexa 488 anti-mouse (#A11029, Invitrogen) for 1 h at 4 °C. Finally, the cells were washed twice with PBS and resuspended in 1 ml of PBS. The Alexa 488 signals of cells were measured by CyFlow Cube 6 (Sysmex). The signals were analyzed with FlowJo™ v10.8 (BD Life Sciences).

### Western blot
Cph1-PM-MDA cells were seeded and grown in 150 mm Petri dishes (Corning) to 80% confluence with 1800 μg/mL G418 at 37 °C and 5% $CO_2$. The cells were then incubated for 24 h under red or far-red light with 5 μM PCB. Before harvesting, cells were treated with pervanadate (50 mM) for 20 min. The plates were put on ice and the cells were triple-washed with ice-cold PBS. The cells were then lifted with a rubber policeman and centrifuged for 5 min at $600 \times g$. The remaining supernatant was discarded, and cells were frozen overnight to initiate cell lysis. Cells were then defrosted and fully lysed in RIPA buffer containing protease inhibitor cocktail (Sigma). The cells were then sonicated for 20 s at 1 s pulses. Protein concentration was then determined by Bradford assay (Thermo Scientific) according to the manufacturer's instructions. The protein samples were boiled at 95 °C for 5 min, and 40 μg protein samples were loaded and run on 10% Bis-tris SDS-PAGE at 100 V for 90 min. Next, proteins were transferred to nitrocellulose membrane (Carl Roth) at 25 V for 90 min. The membranes were blocked with 5% milk in TBST at room temperature for 60 min and probed with 1:1000 mouse-anti-E-cadherin antibody (#14472, Cell Signaling), 1:1000 rabbit-anti-Vimentin antibody (#5741, Cell Signaling), and 1:2000 rabbit-anti-β-actin antibody (#4970, Cell Signaling) overnight at 4 °C. Primary was removed and the membranes were triple-washed with TBST, and 1:1000 HRP-based anti-mouse secondary antibody (#7076, Cell signaling) and 1:1000 HRP-based anti-rabbit secondary antibodies (#7074, Cell Signaling) were used for chemiluminescent evaluation at room temperature for 2 h. Pierce ECL western blotting substrate (Thermo Scientific) was used for the developing process.

### Gene expression analysis
Cph1-PM-MDA cells were seeded at $1.6 \times 10^5$ cells into 45 mm Petri dishes (Corning) and incubated overnight with 5 μM PCB and 1800 μg/mL G418 at 37 °C and 5% $CO_2$. Following, Cph1-PM-MDA cells were illuminated for 24 h at 37 °C and 5% $CO_2$. RNA extraction was performed using TRIzol reagent (Life Technologies), following the manufacturer's protocol. Subsequently, 1 μg of RNA was reverse-transcribed to cDNA with iScript cDNA Synthesis kit (Biorad). Azure Cielo 6 (Azure Biosystem) was used to detect the signal of mRNA expression level with QuiantiNova SYBR Green (Qiagen). Target genes included ZEB1 and FN1 for mesenchymal epithelial transition markers and TEAD1 and CTGF for Yap-associated markers. Primer sequences are provided in Supplementary Table 1. GAPDH was used as housekeeping gene. Relative fold changes were calculated by applying the $2^{(-\Delta\Delta Ct)}$ method[63].

### Statistics and reproducibility
Statistical analyses were conducted to ensure data normality using the Shapiro-Wilk test. When data followed a normal distribution, a two-sample Student's t-test (two-tailed) or a pair-sample Student's t-test (two-tailed) was used for pairwise comparisons; otherwise, a two-sample Mann-Whitney U-test or Kolmogorov-Smirnov test was utilized. For comparisons involving multiple groups, a one-way analysis of variance with Bonferroni's, Scheffe's or Turkey's post-hoc analysis was performed. Statistical analyses were carried out using OriginPro 2020 version 9.7.0.185. The results are presented as mean values with accompanying standard deviations (mean ± SD). The experiments were performed with a minimum of two biologically independent replicates, each with at least two technical replicates. No data were excluded from the analyses.

### Reporting summary
Further information on research design is available in the Nature Portfolio Reporting Summary linked to this article.

## Data availability
All data in this study are available. Primary imaging data of large size will be available for research purposes upon request, provided within 4 weeks and accessible for the next 10 years. Source data are provided with this study. Source data are provided with this paper.

## Code availability
All codes in this paper are available. The codes are also provided in the Supplementary Information.

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

## Acknowledgements

This work was funded by the European Research Council ERC Starting Grant ARTIST (# 757593, S.V.W) and Deutsche Forschungsgemeinschaft (DFG, German Research Foundation,—Project-ID: 386797833— SFB1348/A06, SFB1348/A14, and GA2268/4-1).

## Author contributions

B.M.B., J.P., and S.V.W. designed the experiments. B.M.B and J.P. performed the experiments, analyzed the data, and generated the figures. M.B. and A.D. supported the atomic force microscope measurements. C.T. and M.G. conducted the analysis of single-cell tracks. All authors discussed data and wrote the manuscript.

## Funding

## Competing interests

The authors declare no competing interests.
