## [Transparent Peer Review file · Nature Communications]

Intercellular adhesion boots collective cell migration through elevated membrane tension

Corresponding Author: Professor Seraphine Wegner

Version 0:

Reviewer comments:

Reviewer #1

(Remarks to the Author)

In the present manuscript, Bijonowski and co-authors used MDA-231 investigated the effect of intercellular connections on collective cell migration. They used over-expressed Cph1-MP as photo-switchable artificial intercellular linkers without direct involvement of cytoskeletons, as well as wound-healing assay as a model of collective cell migration. They found that red light, which induces the linking of Cph1-MP, significantly increased the wound closure rate and correlation length of Cph1-MP-MDA cells, indicating that membrane linkage switched the cell migration into a more collective way. They further probed that the artificial linking of cell membrane not only increased the formation of pERM foci but also induced higher membrane tension and more FA formation. Combining with drug treatments, authors attribute the effects to activated PLD2 and mTOR pathway.

Overall, it is a well-done work with significant topics and elegant methods. Especially, they used Cph1-MP as an artificial linker without direct interaction with cytoskeleton. By this way they dissected the effect of cell membrane linkage from intracellular cytoskeleton. But at the same time, critical control experiments are lacking, and some parts of the data is not well presented. Thus, I recommend a major revision of this manuscript.

Major concerns:

- Authors are expected to show, with direct observation, the successful expression of Cph1-MP in MDA cells, by gel (western blot or pull down from lysate), or fluorescent imaging (fluorescent tag or immunostaining), or any other practical method.
- The possibility of direct effect from the light should be ruled out by observing wild-type MDA cells under red and far-red light, at least in Figure 1b and 1c.
- P-values are lacking in several places to claim some statements:
 - o Line 91-92 "As expected, we find significantly faster the wound closure rates in MCF-7 cells compared to MDA cells (Fig. 1b)". But no p value was provided to compare the wound closure rates in MCF-7 cells with MDA cells in Fig. 1b.
 - o Line 194 – 195 "Moreover, the addition of Jasp yielded no significant changes compared to the untreated cells (Figs. 3c, d)" Based on the plot of Fig.3d, it looks like Jasp treatment caused increased wound closure rate under red light and decreased rate under far-red. P-values should be provided if the authors want to claim Jasp does not induce significant changes.
 - o Line 234 – 235 "no significant differences in PA content was observed between Cph1-PM-MDA cells under far-red light and the parent MDA cells." But the plot of Fig. 4a seems to show Cph1-MDA cells under far-red have higher level of PA. P value should be provided to claim they have no significant difference.
- Line 277-278 "SF1670 treatment resulted in an increased coordination of the cells even under far-red light". Statistics of correlation length of Fig. 5c should be calculated to claim this.

Minor concerns:

- The authors should include the equation(s) to calculate correlation length in the methods section.
- Line 627-629: "Phase-contrast micrographs (left) and vector velocity maps (right) depicting cell migratory trajectory of Cph1-PM-MDA cells under red (top) and far-red (bottom) light." The sub fig locations (left, right, top, bottom) are wrongly annotated. Please correct.
- In Sup. Fig. 2c and Sup. Fig. 4, authors should provide a color bar to show the corresponding values of the color codes.
- Line 150 – 151 "we did not observe significant differences in membrane tension in the front vs. rear of the wound for either condition (Supplementary Figs. 2a, b)". Sup. Fig. 2a is the definition of front and rear cells and 2b is the definition of

straightness. Do the authors mean Supplementary Fig. 3a, b?

- Line 273-275 “After wortmannin treatment, the wound closure rate of Cph1-PM-MDA cells under red light significantly decreased and was comparable to cells under far-red light (Fig. 5a)”. Do the authors mean Fig. 5b instead of 5a?
- Line 46 “Regulation at of”. Not standard grammar.

Reviewer #2

(Remarks to the Author)

Review of Bijonowski et al “Intercellular adhesion boots collective cell migration through elevated membrane tension“

This paper presents the development and application of photo-activateable cell-cell adhesion molecules, which is switched on by red light, and inactivated by far-red. I found this to be an interesting and useful approach.

While the applied insights here are limited, I feel the technique is of broad interest to the collective cell migration community and recommend that the article be published once the comments below that endeavor to more fully establish the construct and methodology are addressed.

What are the dynamics of adhesion activation? E.g. how long post activation does it take for collective migration to ensue, and once established, does it persist if the synthetic adhesion is inactivated? This will reveal how sensitive the system is to the adhesions, and the potential role of other intercellular interactions.

I did not see motility analysis of single isolated cells under red or far red light. This is important to show that it is indeed a collective phenomena instead of single-cell changes that also manifest in collective migration.

The story as presented seems to claim that this is entirely driven by membrane tension. While strong evidence of membrane tension is provided, I do not find this well-supported as the sole cause. Although the authors claim that the engineered molecules do not engage cytoskeletal structures, I did not see direct evidence of this. Visualization of stress fibers in activated vs non-activated cases would help. Indeed, Fig 2H&F seems to suggest an accumulation of stress fibers between cells in the activated state.

CytoD is employed, but I find it curious that wound closure is only attenuated by 50-75% and not abrogated. I did not see the concentration/duration used- please ensure this is stated for all reagents.

The authors should investigate intracellular changes. For example in Figure 3 where it is asserted that cell identity has not changed, it would be appropriate minimally to investigate changes in active myosin to determine if the cells are doing more contractile work once adhered to one another, or if it is the existing exerted work which is being optimized.

Reporting the dynamic cellular traction forces would provide a significant boost to the paper to show either a profound reorganization of these forces (intracellular contribution) vs no significant changes with engaged adhesion (solely cell-cell adhesion tension). Alternatively, immunofluorescence of phosphorylated myosin may be a useful comparison.

I found Supp Fig 2 & 4 difficult to understand what is meant.

The writing is poor: some corrected language examples that immediately struck me, but the entire manuscript must be carefully corrected:

Line 33 “For instance”

line 35 “mechanical link between cells that not only acts as structural elements but also activates ”

The writing is also a bit cumbersome and should be streamlined.

Line 309 is too strong of a statement “we ensured that the differences in migration only arise from the connections between the cells and not from some other genetic or environmental factors. “

Reviewer #3

(Remarks to the Author)

The authors of “Intercellular adhesion boots collective cell migration through elevated membrane tension” use a recently characterised photoswitchable tool that modulates levels of cell-cell adhesion to understand whether adhesion-dependent effects on the plasma membrane are involved in directly controlling collective cell migration. The critical aspect of this tool is that it allows for the direct manipulation of cell-cell adhesion without confounding effects due to intracellular signaling from the cytoplasmic domain of adhesion molecules. While the work focuses on a single cell line, the results are potentially broadly relevant as membrane tension is increasingly appreciated to be an important signaling mediator. While the results, and the novel approach, are interesting, there are a few details regarding their hypothesised mechanisms downstream of membrane tension leading to collective migration that I believe the authors should address prior to publication. Additionally, the photoswitchable tool is potentially very interesting for the community and it is important they characterise it and explain its use properly in the methods. The data presentation is also not great in my opinion and analytical methods are also incomplete. It is therefore a little difficult to assess the data.

- In the introduction the authors state that “signaling cues that originate from within the plasma membrane for collective cell dynamics has largely been overlooked”. Could they provide some background about these ‘cues’? Why should we be interested in these cues? I feel like there is missing background and the importance is lost on the non-expert.
- The Cph1-PM tool that the authors somewhat characterised in a previous publication to control cell-cell adhesion is interesting. There needs to be more detail on the methods regarding how they use this tool. For example, how long do they have to illuminate in the red wavelength in order to induce the increase in cell-cell adhesion?
- The authors use the Cph1-PM in a binary fashion to increase cell-cell adhesion. Is it possible to alter the intensity of the red light to examine subtle changes in the amount of cell-cell adhesion?
- What are ‘directional rate vectors’? I am guessing this is just the vectors highlighting cell motion?
- They say that “the majority of motion vectors were directed towards the wound space”. This could easily be quantified (e.g. rose plots) and compared under different illumination. Was the orientation actually ‘random’ under far red?? I doubt it considering that the wounds eventually close.
- Please give some methods on the analysis of correlation length.
- Please give methods on the analysis of persistence and straightness index. What was the deltaT used for the analysis. The length over which they are quantifying persistence needs to be the same.
- Line 91-92. “As expected, we find significantly faster the wound closure rates in MCF-7 cells compared to MDA cells (Fig. 1b).” but the graph did not show a significant difference.
- They state that “Cph1-PM mediated transcellular adhesion augments plasma membrane tension”. Do they know that it is definitely transcellular? Proteins can interact in cis as well. Could they look at individual cells not in contact with neighbors to make sure that they don’t see an increase in plasma membrane tension?
- The authors cite previous work that reports that the increase in friction between individual cells slows down cell speed while increasing collectivity (ref 26). However, I think that this is an overinterpretation of this paper. The work was performed in endothelial cells treated with Fgf while knocking down cadherins. There are not claims about the direct effects of friction in this paper.
- On line 137 they state that they focused on a single cell-cell interface they observe an increase in GP. What does ‘a single cell-cell interface’ mean? And is it only the cell interfaces in contact with a neighbor that they observe an increase in ordering (i.e. do they observe a similar increase for cells at the leading edge which have fewer contacts with neighbors?)?
- In figure 2a and b, please show the cells along with the heatmap. I am guessing that this is in an unwounded monolayer? What does the heatmap look like when wounded? Is there variation in plasma membrane tension from the leading edge into the interior? It is also not possible to visibly see a difference in the images that they display.
- They measure the PM tension under red light illumination. How quickly does this increase happen after illumination? Related to the comment above, does the intensity of the light affect the tension? Being able to modulate the amount of adhesion and PM tension could potentially be informative.
- On line 153 they state that “Cph1-PM mediated changes in membrane tension affect membrane to cortex attachment and formation of focal adhesions”. This suggests that increase in tension happens before changes in the cortex and the formation of focal adhesions. Is it possible to address the dynamics of these changes? I understand it may be difficult to address the timecourse of events. If so, the authors need to soften some of their statements which suggest cause and effect.
- The authors observe an increase in ERM phosphorylation after illumination. Do they still observe an increase in tension when removing the membrane cortex crosslinking? How much of the tension increase is related to the increased activity of Cph1 and how much of it is related to downstream signaling?
- The authors analysed changes in E-cadherin after illumination. However, what about other cadherins, such as N-cadherin? Presumably, if these cells are mesenchymal then they should still have N-cad. And just because they do not observe a change in levels of cadherins, it doesn’t mean that adhesion proteins may not also be involved in the response.
- They chemically perturb the actin network and examine wound closure. However, does perturbing actin also effect membrane tension after red light illumination?
- There is no reference to movie 4. And in line 265 they incorrectly reference video 4 when they should instead reference video 5.
- In movie 4 and 5 the cell densities look different in the far red vs red treatments. The red cell densities are higher. Can we be sure that the differences in migration are not related to differences in starting cell densities?
- They directly affected membrane tension through the addition of oleic acid (OA), which led to an increase in migration speed without an increase in cell correlation. Doesn’t this suggest that the increase in speed can be uncoupled from the increase in cell coordination in response to an increase in membrane tension? The following analysis of PI3K/mTor appears to suggest that increased speed and cell coordination are in response to a single pathway downstream of the increase in membrane tension. But maybe membrane tension has pleiotropic effects on the cell.
- Inhibition of PTEN with SF1670 resulted in an increased coordination of the cells under control conditions. They also state that SF1670 increased the rate of wound closure in control cells. Is this counter to their initial argument that increased collective interactions should slow down migration? Any idea how PTEN inhibition is doing this?
- I am not sure I understand the interpretation of mTor inhibition with rapamycin. They state that the rate of wound closure of Cph1/red cells decreased after rapamycin, arguing that inhibition of the mTor pathway selectively suppressed the collective behavior of the cells. What is the relationship of wound closure speed and collective motion?? And in figure 5 they do not quantify cell collectivity, only speed.

Figure comments.

- Fig 1b (and 3d, 3h, 3i, 4a, 4e, 4i, 5b): please display as box and whiskers plus data points or datapoints only (with error bars) - not bar plot.
- Fig 1c (and 3e, 4f, 4j): display single data points overlaid to box and whiskers plot

- Fig 1d (and 3b, 4c, 4d, 4h): what is the PIV measured on? Time 3h vs time 0? Are velocity vectors normalised and how? What min/max represent in the colour bar? No details in caption, please fix
- Fig 1e: what is a directional rate vector? Are these calculated from the PIV or separately? Is this a single particle tracking as suggested by the colour bar legend? No details in the methods.
- Fig 1f: what is the value assigned to one cell diameter? What is a 'straightness index' and how is it calculated? Again, no details in the methods, please fix.
- Fig 1g: What is a 'straightness index' and how is it calculated? Again, no details in the methods, please fix. Please display single data points overlaid to box and whiskers plot
- Fig 1i: display single data points overlaid to box and whiskers plot. Figure panel not mentioned in the text.
- Fig 2a,b: how is the normalized general polarization calculated/measured?
- Fig 2c: In the methods they state "contact time was varied between 100 ms to 10 s, aiming at maximizing the probability of extruding single tethers" – this is a big time range, can they please elaborate further? Is this standard for this assay?
- Fig 2g (and 3g, 4a): how was this quantified?
- Fig 2i: how was cell area quantified? Was actin used as a proxy?
- Fig 2j-l: how was FA area quantified?
- Fig2: check panel letters in legend
- Image analysis methods lack details. Please describe the analysis better and provide strategies and chosen parameters for all the software tools used, keeping in mind a reader should be able to replicate your results. For example, "For wound healing, images were analyzed with ImageJ (National Institute of Health, USA) to determine the area of the wound at each time point": how was the analysis performed? Which plugin tools were used if any?

Minor comment

Can you please make sure that the order of figure panels follows the text and the alphabetical order?

Version 1:

Reviewer comments:

Reviewer #1

(Remarks to the Author)

The authors addressed all my concerns and now I support the publication of this manuscript.

Reviewer #2

(Remarks to the Author)

The authors have made substantial improvements to their manuscript and have addressed my concerns. I recommend the manuscript be published.

Reviewer #3

(Remarks to the Author)

The authors have sufficiently dealt with my comments.

REVIEWER COMMENTS

Reviewer #1 (Remarks to the Author):

In the present manuscript, Bijonowski and co-authors used MDA-231 investigated the effect of intercellular connections on collective cell migration. They used over-expressed Cph1-MP as photo-switchable artificial intercellular linkers without direct involvement of cytoskeletons, as well as wound-healing assay as a model of collective cell migration. They found that red light, which induces the linking of Cph1-MP, significantly increased the wound closure rate and correlation length of Cph1-MP-MDA cells, indicating that membrane linkage switched the cell migration into a more collective way. They further probed that the artificial linking of cell membrane not only increased the formation of pERM foci but also induced higher membrane tension and more FA formation. Combining with drug treatments, authors attribute the effects to activated PLD2 and mTOR pathway.

Overall, it is a well-done work with significant topics and elegant methods. Especially, they used Cph1-MP as an artificial linker without direct interaction with cytoskeleton. By this way they dissected the effect of cell membrane linkage from intracellular cytoskeleton. But at the same time, critical control experiments are lacking, and some parts of the data is not well presented. Thus, I recommend a major revision of this manuscript.

Response: We appreciate the reviewer's assessment and have addressed all the concerns as detailed below.

Major concerns:

Comment:

Authors are expected to show, with direct observation, the successful expression of Cph1-MP in MDA cells, by gel (western blot or pull down from lysate), or fluorescent imaging (fluorescent tag or immunostaining), or any other practical method.

Response:

To generate a stable monoclonal Cph1-PM-MDA cell line, we initially sorted the cells based on the GFP signal using FACS. As suggested, we further confirmed the expression of the Cph1 on the cell surface using flow cytometry and immunofluorescence imaging of the Myc-tag on the construct (Supplementary Figs. 1a, b). We added the following sentence:

“Successful expression of Cph1 on the cell surface was confirmed by immunostaining unpermeabilized cells for the Myc-tag and subsequently analyzing them with flow cytometry and confocal microscopy (Supplementary Fig. 1a, b).”

Comment:

The possibility of direct effect from the light should be ruled out by observing wild-type MDA cells under red and far-red light, at least in Figure 1b and 1c.

Response:

Following the suggestion, we conducted the wound healing assay for MDA cells in the dark as well as under red and far-red light illumination. We found no difference in wound closure rate and correlation length in all three conditions, ruling out a possible effect of the light illumination (Supplementary Fig. 1g, h). We added the following sentence:

“Finally, to ensure that light illumination did not influence cell migration, we performed a wound-healing assay under various light conditions. We found that the wound-healing rate and correlation length remained unchanged under red and far-red light compared with those under

darkness, suggesting that illumination itself had no effect on cell migration (Supplementary Fig. 1g, h)..”

Comment:

P-values are lacking in several places to claim some statements: Line 91-92 “As expected, we find significantly faster the wound closure rates in MCF-7 cells compared to MDA cells (Fig. 1b) “. But no p value was provided to compare the wound closure rates in MCF-7 cells with MDA cells in Fig. 1b.

Response:

We corrected the statement as the difference of MCF7 and MDA cells is not in the wound closure rate but in the correlation length. We thank the reviewer for spotting this error.

“We observed similar wound closure rates in MCF-7 and MDA cells (Fig. 1b). However, the correlation length was higher in MCF-7 cells (Fig. 1c), indicating a greater degree of coordinated cell movement in this cell line.”

Comment:

Line 194 – 195 “Moreover, the addition of Jasp yielded no significant changes compared to the untreated cells (Figs. 3c, d)” Based on the plot of Fig.3d, it looks like Jasp treatment caused increased wound closure rate under red light and decreased rate under far-red. P-values should be provided if the authors want to claim Jasp does not induce significant changes.

Response:

We carefully evaluated the p-values for the wound healing rate between the control and Jasp-treated cells under red and far-red light. A statistically significant difference was observed under red light ($p = 0.0028$), but not under far-red light ($p = 0.21$), in terms of wound closure rate. However, the correlation length increased with Jasp treatment under both illumination conditions. We added the p values to the graphs and revised the manuscript as follows:

“Addition of Jasp under far-red light yielded no significant changes compared with untreated cells, but increased the wound-healing rate under red light (Fig. 3c, d). At the same time, the correlation length increased in Jasp-treated cells under both red and far-red light, but the difference between the two illumination conditions remained stable (Fig. 3e).”

Comment:

Line 234 – 235 “no significant differences in PA content was observed between Cph1-PM-MDA cells under far-red light and the parent MDA cells.” But the plot of Fig. 4a seems to show Chp1-MDA cells under far-red have higher level of PA. P value should be provided to claim they have no significant difference.

Response:

The statistical analysis of the PA contents between far-red and parent MDA cells shows no significance ($p=0.65$) and the p value has been added to Fig 4a.

Comment:

Line 277-278 “SF1670 treatment resulted in an increased coordination of the cells even under far-red light”. Statistics of correlation length of Fig. 5c should be calculated to claim this.

Response:

We added the correlation length analysis as Fig 5c to support this claim. We also edited the sentence as follows:

“At the same time, the correlation length decreased with both drugs, regardless of light illuminations (Fig. 5c).”

Minor concerns

Comment:

The authors should include the equation(s) to calculate correlation length in the methods section.

Response:

*As suggested, we added the MATLAB code used to compute the correlation length as **Supplementary Code 1**.*

Comment:

Line 627-629: “Phase-contrast micrographs (left) and vector velocity maps (right) depicting cell migratory trajectory of Cph1-PM-MDA cells under red (top) and far-red (bottom) light.” The sub fig locations (left, right, top, bottom) are wrongly annotated. Please correct.

Response:

We thank the reviewer for pointing out this mistake, which we corrected now stating: “Phase-contrast micrographs (top) and velocity vector maps (bottom) for Cph1-PM-MDA cells under red and far-red light on addition of (b) CytoD and (c) Jasp.”

Comment:

In Sup. Fig. 2c and Sup. Fig. 4, authors should provide a color bar to show the corresponding values of the color codes.

Response:

*Following the suggestion, we added a color bar with the corresponding values to **Supplementary Fig. 2 and 8**.*

Comment:

Line 150 – 151 “we did not observe significant differences in membrane tension in the front vs. rear of the wound for either condition (Supplementary Figs. 2a, b)”. Sup. Fig. 2a is the definition of front and rear cells and 2b is the definition of straightness. Do the authors mean Supplementary Fig. 3a, b?

Response:

We corrected the numbering of the figure.

Comment:

Line 273-275 “After wortmannin treatment, the wound closure rate of Cph1-PM-MDA cells under red light significantly decreased and was comparable to cells under far-red light (Fig. 5a)”. Do the authors mean Fig. 5b instead of 5a?

Response:

We corrected the numbering of the figure.

Comment:

Line 46 “Regulation at of”. Not standard grammar.

Response:

We corrected the grammatical mistake.

Reviewer #2 (Remarks to the Author):

Review of Bijonowski et al “Intercellular adhesion boots collective cell migration through elevated membrane tension”

This paper presents the development and application of photo-activateable cell-cell adhesion molecules, which is switched on by red light, and inactivated by far-red. I found this to be an interesting and useful approach.

While the applied insights here are limited, I feel the technique is of broad interest to the collective cell migration community and recommend that the article be published once the comments below that endeavor to more fully establish the construct and methodology are addressed.

Response: We appreciate the reviewer’s assessment and have addressed all the concerns as detailed below.

Comment:

What are the dynamics of adhesion activation? E.g. how long post activation does it take for collective migration to ensue, and once established, does it persist if the synthetic adhesion is inactivated? This will reveal how sensitive the system is to the adhesions, and the potential role of other intercellular interactions.

Response:

The question of dynamics at different length scales is a topic of significant interest. To address this, we conducted additional experiments and included a detailed discussion in the manuscript.

*To investigate how long it takes to establish or lose collective migration, we performed the wound healing assay by exposing Cph1-PM-MDA cells to one type of illumination (red or far-red light) for 6 hours, followed by 12 hours of exposure to the opposite type of illumination. We then analyzed the wound closure rate and correlation length at different time points (**Supplementary Fig. 3**). As described in the methods section, the first 4 hours of the wound healing were excluded from the analysis due to the high variability in cell migration caused by the initial wounding. Our results show that the wound healing rate decreased immediately after switching cells after 6 hours from red to far-red light illumination (**Supplementary Figs 3a, b**). Additionally, the correlation length also decreased after far-red light exposure compared to red illumination (**Supplementary Fig 3c**), indicating a rapid reversal of Cph1 based adhesions and collective cell movement following far-red light exposure. In contrast, when the cells were switched from far-red to red light after 6 hours, neither the wound closure rate nor the correlation length increased (**Supplementary Figs 3d, e, f**). This is because, during the 6 hours of far-red light illumination, individual cells had already migrated apart and moved randomly into the wound. For the Cph1 based cell-cell adhesions to form, turning on red light is not sufficient; the rate-limiting step in this context is cells coming into proximity. We have added the following text to the manuscript to elaborate on this point:*

*“The photoregulation of the Cph1-based cell-cell adhesions enable dynamic and reversible control over these adhesions and the resulting cell migration. To investigate this, we examined whether switching illumination from red light to far-red light, or vice versa, would alter the migration behavior. In the wound-healing assay, we illuminated Cph1-PM-MDA cells with red light for the first 6 hours, followed by far-red light for the next 12 hours (**Supplementary Fig. 4a, Supplementary Movie 4**). We observed that the wound-healing rate began to decrease*

immediately after switching from red to far-red light. Moreover, the average wound-healing rate and correlation length both decreased following the change in illumination (Supplementary Fig. 4b, c). On the other hand, when the cells were exposed to far-red light for 6 hours followed by red light for 12 hours, the wound-healing rate did not significantly change after switching the illumination (Supplementary Fig. 4d, e, Supplementary Movie 5). At the same time, the correlation length did not increase under red light illumination until the end of the experiment (Supplementary Fig. 4f), indicating a lack of collective and coordinated movement. We reason that the cells had already dispersed during the initial 6 hours of far-red light exposure. Although Cph1 at the molecular level is activated within milliseconds of red light illumination 34, cells must be in close proximity for cell-cell adhesions to form and for collective cell migration to emerge. Likewise, Cph1 is also deactivated within milliseconds of far-red light activation, but it takes longer for cells to separate from each other."

Comment:

I did not see motility analysis of single isolated cells under red or far red light. This is important to show that it is indeed a collective phenomena instead of single-cell changes that also manifest in collective migration.

Response:

To address this question, we seeded Cph1-PM-MDA cells at low confluence and analyzed the speed, shape and cell division of individual cells under red and far-red light illumination. Our analysis revealed no significant difference between the two illumination conditions (Supplementary Figs. 3a-f). This finding demonstrates that the observed collective cell migration and the associated signaling are collective phenomena, rather than effects on individual cell behavior. Following comment was added to the manuscript:

"Finally, to account for the possibility that photoactivation by itself changed cell dynamics, we cultured Cph1-PM-MDA cells at low density and tracked individual cells on exposure to red and far-red light. We found no differences in cell shape, cell division, or cell speed (Supplementary Fig. 3a-f). Collectively, these experiments suggest that the observed changes in collective cell dynamics are due to changes in transcellular and not intracellular dynamics."

Comment:

The story as presented seems to claim that this is entirely driven by membrane tension. While strong evidence of membrane tension is provided, I do not find this well-supported as the sole cause. Although the authors claim that the engineered molecules do not engage cytoskeletal structures, I did not see direct evidence of this. Visualization of stress fibers in activated vs non-activated cases would help. Indeed, Fig 2H&F seems to suggest an accumulation of stress fibers between cells in the activated state.

Response:

We thank the reviewer for bringing up this point. We clarified in the design of Cph1-PM that there is no direct molecular link between the adhesion molecule Cph1-PM and the actin cytoskeleton as Cph1-PM lacks the intracellular tail to directly bind and/or signal to the actin cytoskeleton.

"Unlike classical cadherins, Cph1-PM lacks an intracellular domain for adaptor proteins to bind to, preventing direct linkage of these artificial adhesions to the actin cytoskeleton."

As correctly stated by the referee, we observe that the cytoskeleton is changed by the artificial cell-cell adhesions under red light, and there are stress fibers along the edges. We link this to the increase in pERM levels, connecting the membrane to the actin cortex. While a detailed understanding of the underlying signaling that drives these changes would be desirable, we consider this beyond the scope of this study: The principal observation of this manuscript is the activation of a signaling circuit through changes in membrane tension. We do, however, agree

with the referee that additional effects cannot be excluded. We clarify this point by adding the following sentence to the discussion of the revised manuscript:

“Notably, although this study established a signaling cascade of Cph1-PM through tension-dependent activation of TOR signaling, it does not exclude the possibility that additional signaling cascades are activated.”

Comment:

CytoD is employed, but I find it curious that wound closure is only attenuated by 50-75% and not abrogated. I did not see the concentration/duration used- please ensure this is stated for all reagents.

Response:

The CytoD concentration was picked such that it was not toxic, while preventing cell migration. In the MTT test we observed toxicity at concentrations above 200 nM and therefore used 100 nM to prevent actin polymerization (Supplementary Fig. 7a-c). We specified the concentration and duration in the revised manuscript and the methods as follows:

“Because of the strong cell toxicity of CytoD, its concentration was optimized to remain below toxicity while still preventing actin polymerization (Supplementary Fig. 7a-c).”

Comment:

The authors should investigate intracellular changes. For example in Figure 3 where it is asserted that cell identity has not changed, it would be appropriate minimally to investigate changes in active myosin to determine if the cells are doing more contractile work once adhered to one another, or if it is the existing exerted work which is being optimized.

Reporting the dynamic cellular traction forces would provide a significant boost to the paper to show either a profound reorganization of these forces (intracellular contribution) vs no significant changes with engaged adhesion (solely cell-cell adhesion tension). Alternatively, immunofluorescence of phosphorylated myosin may be a useful comparison.

Response:

We agree that dynamic cellular traction forces would certainly be valuable. Yet, as the reviewer is certainly aware, traction force microscopy requires specialized expertise and is not a readily available and accessible technique. We choose the proposed alternative and looked at the phosphorylated myosin. Performing immunostaining for phosphorylated myosin, we observed an increase in pMyosin, especially in the leading edge under red light (Supplementary Fig. 6e). Since contractility indicates the direction in which the cells move, this supports the observation that cells in the wound region showed higher motility under red light compared to far-red light. We added the following sentences:

“At the same time, we observed an accumulation of actin stress fibers along the cell edge under red light, as well as an increase in phosphorylated myosin, particularly in cells at the wound edge, suggesting enhanced contractility (Supplementary Fig. 6e).”

Comment:

I found Supp Fig 2 & 4 difficult to understand what is meant.

Response:

Following the referee’s recommendation, the figure legends were extended. We further added the following explanation to the methods section to better explain the image analysis pipeline:

“To analyze the migration pattern during wound closure, single-cell tracking was performed using the TrackMate Plugin (v7.11.1,) in ImageJ (1.54f). For particle detection and linking, a Laplacian

of Gaussian (LoG) detector and a Simple Linear Assignment Problem Tracker were used, respectively. Detection of the wound edge and the analysis of migration parameters including cells speed, straightness index and the orientation of migration were performed in MATLAB (R2020b).

To distinguish between cells located close to the wound edge and cells further back in the field, a line parallel to the wound edge was drawn with a distance of 60 μm , which is approximately the size of three nuclei. If the center of a nuclei was located within this region, a cell was considered to be migrating at the front. Trajectories were cut off before cells from opposite sides of the wound were touching each other, trajectories shorter than four steps were excluded from analysis.

The straightness index was determined as the ratio between the net displacement and the total distance traveled, ranging from 0 (= movement back to the origin) to 1 (=straight trajectory). The cell speed represents the median speed of each cell in the respective time interval. As an indicator of directionality, the angle of migration in relation to the wound edge is used, referred to as the orientation angle. Angles around zero degree indicate migration towards the open space, the wound edge is here considered to be a straight vertical line."

Comment:

The writing is poor: some corrected language examples that immediately struck me, but the entire manuscript must be carefully corrected: Line 33 "For instance"

Response:

We improved the language of the manuscript.

Comment:

line 35 "mechanical link between cells that not only acts as structural elements but also activates". The writing is also a bit cumbersome and should be streamlined.

Response:

We rephrased the sentence as follows:

"Hence, cadherins both mechanically connect cells as structural elements and activate intercellular signaling circuits that regulate cytoskeletal dynamics and gene expression."

Comment:

Line 309 is too strong of a statement "we ensured that the differences in migration only arise from the connections between the cells and not from some other genetic or environmental factors. "

Response:

We moderated the statement as follows:

"By using the artificial photoswitchable Cph1-PM-based cell-cell connections, we aimed to minimize the influence of other genetic or environmental factors, ensuring that the observed differences in migration were primarily due to the connections between the cells."

Reviewer #3 (Remarks to the Author):

The authors of "Intercellular adhesion boots collective cell migration through elevated membrane tension" use a recently characterised photoswitchable tool that modulates levels of cell-cell adhesion to understand whether adhesion-dependent effects on the plasma membrane are involved in directly controlling collective cell migration. The critical aspect of this tool is that it allows for the direct manipulation of cell-cell adhesion without confounding effects due to intracellular signaling from the cytoplasmic domain of adhesion molecules. While the work focuses on a single cell line, the results

are potentially broadly relevant as membrane tension is increasingly appreciated to be an important signaling mediator. While the results, and the novel approach, are interesting, there are a few details regarding their hypothesised mechanisms downstream of membrane tension leading to collective migration that I believe the authors should address prior to publication. Additionally, the photoswitchable tool is potentially very interesting for the community and it is important they characterise it and explain its use properly in the methods. The data presentation is also not great in my opinion and analytical methods are also incomplete. It is therefore a little difficult to assess the data.

Response: We appreciate the reviewer's assessment. We have addressed all the concerns and added a more detailed description of the methods as detailed below.

Comment:

In the introduction the authors state that “signaling cues that originate from within the plasma membrane for collective cell dynamics has largely been overlooked”. Could they provide some background about these ‘cues’? Why should we be interested in these cues? I feel like there is missing background and the importance is lost on the non-expert.

Response:

We have extended the introduction for the non-experts on the state of the art.

“In contrast to signaling circuits emanating from protein complexes that link the cytoskeletons of adjacent cells, the contribution of signaling cues originating directly from the plasma membrane in collective cell dynamics has been largely overlooked. The membrane is often regarded as a passive structure whose primary role is to separate the cell's exterior from its interior, thereby enabling signal transduction via transmembrane protein complexes. However, recent studies challenge this view. In single cells, changes in membrane properties have been shown to significantly influence leading edge dynamics^{15, 16, 17} and migration behavior^{18, 19, 20}. Notably, these alterations in membrane properties are accomplished not only by direct mechano-chemical feedback loops, but also by intracellular signaling and changes in gene expression^{21, 22}. Consequently, changes at the plasma membrane not only yield short-term changes in cell, but also have long-term effects^{23, 24}. Collectively, these studies suggest that protein-independent signaling cascades that originate from within the plasma membrane are suitable for initiating sustained changes in cell dynamics. Despite this, although it is evident that membrane mechanics regulate the dynamics of individual cells, the contribution of plasma membrane-derived signals to collective cell migration – independent of the direct link between adhesions and cytoskeleton and the associated biochemical signaling cascades – remains unclear. A significant challenge in addressing this question is the lack of tools capable of dissecting the mechanical cues at the plasma membrane from the mechano-transduction through cell-cell adhesion molecules to the cytoskeleton and biochemical signals. For example, the truncation of the cytoplasmic catenin-binding domains renders E-cadherin nonfunctional, as the membrane-anchored extracellular domain alone is insufficient to sustain intercellular cell-cell adhesions^{13, 14}.”

Comment:

The Cph1-PM tool that the authors somewhat characterised in a previous publication to control cell-cell adhesion is interesting. There needs to be more detail on the methods regarding how they use this tool. For example, how long do they have to illuminate in the red wavelength in order to induce the increase in cell-cell adhesion?

Response:

At the molecular level, Cph1 activation and deactivation with red and far-red light, respectively, happens within ms. However, the formation of cell-cell adhesions usually takes longer as cells have to come in proximity, which depends on their density and mobility. We addressed and

discussed this aspect in the revised manuscript in experiments in which we changed illumination during the wound healing assay.

We specify the illumination time in the description of the experiments and in the methods. These can be for 10-15 min as it is the case for membrane tension measurements with Pro12A and AFM or for 16 hours as it was the case during wound healing assays:

*“The photoregulation of the Cph1-based cell-cell adhesions enable dynamic and reversible control over these adhesions and the resulting cell migration. To investigate this, we examined whether switching illumination from red light to far-red light, or vice versa, would alter the migration behavior. In the wound-healing assay, we illuminated Cph1-PM-MDA cells with red light for the first 6 hours, followed by far-red light for the next 12 hours (**Supplementary Fig. 4a, Supplementary Movie 4**). We observed that the wound-healing rate began to decrease immediately after switching from red to far-red light. Moreover, the average wound-healing rate and correlation length both decreased following the change in illumination (**Supplementary Fig. 4b, c**). On the other hand, when the cells were exposed to far-red light for 6 hours followed by red light for 12 hours, the wound-healing rate did not significantly change after switching the illumination (**Supplementary Fig. 4d, e, Supplementary Movie 5**). At the same time, the correlation length did not increase under red light illumination until the end of the experiment (**Supplementary Fig. 4f**), indicating a lack of collective and coordinated movement. We reason that the cells had already dispersed during the initial 6 hours of far-red light exposure. Although Cph1 at the molecular level is activated within milliseconds of red light illumination³⁴, cells must be in close proximity for cell-cell adhesions to form and for collective cell migration to emerge. Likewise, Cph1 is also deactivated within milliseconds of far-red light activation, but it takes longer for cells to separate from each other.”*

Comment:

The authors use the Cph1-PM in a binary fashion to increase cell-cell adhesion. Is it possible to alter the intensity of the red light to examine subtle changes in the amount of cell-cell adhesion?

Response:

*It is possible to partially activate Cph1-PM based cell-cell adhesions using different intensities of red light. To determine the light sensitivity profile of Cph1-PM, we conducted the cell aggregation assay in suspension culture under different intensities of red light. At 1 $\mu\text{W}/\text{cm}^2$ the cell aggregation remained as low as in the dark and at intensities higher than 15 $\mu\text{W}/\text{cm}^2$ a maximum aggregation ratio was achieved (**Supplementary Fig. 6a, b**). Throughout the study we used 1440 $\mu\text{W}/\text{cm}^2$, which assures full activation. We added this point to the manuscript.*

*“As activation of Cph1-PM-based cell-cell adhesions can be modulated by varying red light intensities (**Supplementary Fig. 6a, b**), we consistently used an intensity of 1440 $\mu\text{W}/\text{cm}^2$ throughout the whole study, which is significantly higher than the 15 $\mu\text{W}/\text{cm}^2$ required for full activation.”*

Comment:

What are ‘directional rate vectors’? I am guessing this is just the vectors highlighting cell motion?

Response:

Indeed, this was a misuse of a term. We replaced this with “velocity vector”.

Comment:

They say that “the majority of motion vectors were directed towards the wound space”. This could easily be quantified (e.g. rose plots) and compared under different illumination. Was the

orientation actually 'random' under far red?? I doubt it considering that the wounds eventually close.

Response:

*Following the suggestion, we added the rose plots for the cells under red and far-red light. Of course, in both cases there is a strong preference to move into the wound but under red light more cells moved more perpendicular to the wound (i.e., 0° angle) than under far-red light (**Supplementary Figs. 1d, e**). We modified the statement as follows:*

*“Velocity vector analysis of cell migration revealed that, under red light, a greater proportion of vectors were oriented perpendicular to the wound edge compared with those observed under far-red light (**Supplementary Figs. 1d, e**).”*

Comment:

Please give some methods on the analysis of correlation length.

Response:

*Following the suggestion, we added the Matlab code for the analysis of the correlation length as **Supplementary Code 1**.*

Comment:

Please give methods on the analysis of persistence and straightness index. What was the deltaT used for the analysis. The length over which they are quantifying persistence needs to be the same.

Response:

We thank the referee for pointing out these omissions. Following section was added to the methods section to clarify this point:

“To analyze the migration pattern during wound closure, single-cell tracking was performed using the TrackMate Plugin (v7.11.1, Ershov et al.) in ImageJ (1.54f). For particle detection and linking, a Laplacian of Gaussian (LoG) detector and a Simple Linear Assignment Problem Tracker were used, respectively. Detection of the wound edge and the analysis of migration parameters including cells speed, straightness index and the orientation of migration were performed in MATLAB (R2020b).

To distinguish between cells located close to the wound edge and cells further back in the field, a line parallel to the wound edge was drawn with a distance of 60 μm, which is approximately the size of three nuclei. If the center of a nuclei was located within this region, a cell was considered to be migrating at the front. Trajectories were cut off before cells from opposite sides of the wound were touching each other, trajectories shorter than four steps were excluded from analysis.

The straightness index was determined as the ratio between the net displacement and the total distance traveled, ranging from 0 (= movement back to the origin) to 1 (=straight trajectory). The cell speed represents the median speed of each cell in the respective time interval. As an indicator of directionality, the angle of migration in relation to the wound edge is used, referred to as the orientation angle. Angles around zero degree indicate migration towards the open space, the wound edge is here considered to be a straight vertical line.”

Comment:

Line 91-92. “As expected, we find significantly faster the wound closure rates in MCF-7 cells compared to MDA cells (Fig. 1b).” but the graph did not show a significant difference.

Response:

We appreciate the reviewer's attention to this error in our previous statement. We corrected this statement as follows:

"We observed similar wound closure rates in MCF-7 and MDA cells (Fig. 1b). However, the correlation length was higher in MCF-7 cells (Fig. 1c), indicating a greater degree of coordinated cell movement in this cell line."

Comment:

They state that "Cph1-PM mediated transcellular adhesion augments plasma membrane tension". Do they know that it is definitely transcellular? Proteins can interact in cis as well. Could they look at individual cells not in contact with neighbors to make sure that they don't see an increase in plasma membrane tension?

Response:

The crystal structure of Cph1 reveals antiparallel dimer formation, which, when the protein is immobilized on the cell surface, is expected to result in trans interactions rather than cis interactions. Additionally, increased membrane tension is known to enhance cell motility in single cells. However, we do not observe an increase in single-cell migration speed for Cph1-PM-MDA cells under red light compared to far-red light. This finding further supports the notion that Cph1 does not interact in cis on the membrane and that such interactions do not contribute to increased membrane tension. We added following sentence to the manuscript clarifying the interaction of Cph1 in trans:

"Cph1 forms antiparallel dimers, as observed in its crystal structure²⁷, and therefore, when expressed on cell surfaces, it forms interactions in trans rather than in cis."

Comment:

The authors cite previous work that reports that the increase in friction between individual cells slows down cell speed while increasing collectivity (ref 26). However, I think that this is an overinterpretation of this paper. The work was performed in endothelial cells treated with Fgf while knocking down cadherins. There are not claims about the direct effects of friction in this paper.

Response:

We thank the referee for pointing out this overstatement. We changed the text as follows:

"Several studies indicate that the strength of intercellular connections significantly affects collective cell dynamics (e.g., glass transition^{30, 31}, plithotaxis^{32, 33}), whereby "fluidification" resulting from reduced intercellular friction is generally associated with increased cell speed and a decrease in correlation length."

Comment:

On line 137 they state that they focused on a single cell-cell interface they observe an increase in GP. What does 'a single cell-cell interface' mean? And is it only the cell interfaces in contact with a neighbor that they observe an increase in ordering (i.e. do they observe a similar increase for cells at the leading edge which have fewer contacts with neighbors?)?

In figure 2a and b, please show the cells along with the heatmap. I am guessing that this is in an unwounded monolayer? What does the heatmap look like when wounded? Is there variation in plasma membrane tension from the leading edge into the interior? It is also not possible to visibly see a difference in the images that they display.

Response:

*A single cell-cell interface means that we find two cells neighboring each other and do the ratiometric analysis of just the contact area. To prevent confusion, we exchanged the word “single cell-cell interface” to “the contact area between two cells”. We performed a similar measurement between cells at the wound edge (**Supplementary Fig. 5a, b**) and still observe a higher GP under red light than under far-red light. This finding is in agreement with the AFM results, in which we observed no difference in membrane tension between leader and follower cells in the same group. Yet, the dynamic range of the GP measurements is not very high and the sample-to-sample variation is too high to quantitatively say if the change in GP directly correlates with the number of neighbors or the light intensity (see comment below).*

*We provide an overview fluorescence image in **Fig. 2a**, where the cell-cell contact areas are visible and it is obvious that these are cells within a confluent layer of cells.*

*“Focusing on contact areas between two cells in a confluent layer, we observed an increase in GP when Cph1-PM was activated with red light for 15 minutes (**Fig. 2a**). In contrast, when the cells were first exposed to red light for 15 minutes and then to far-red light for 15 minutes, to inactivate Cph1-PM-based adhesions, the GP values decreased (**Fig. 2b**). The changes in GP were fast and first responses were observed within 5 to 10 minutes at the time resolution of the imaging (**Supplementary Fig. 5a**). Moreover, similar trends in GP values were also observed for cells at the wound edges when switching from red to far-red light and vice versa (**Supplementary Fig. 5b**).”*

Comment:

They measure the PM tension under red light illumination. How quickly does this increase happen after illumination? Related to the comment above, does the intensity of the light affect the tension? Being able to modulate the amount of adhesion and PM tension could potentially be informative.

Response:

*In time dependent the GP measurements, when switching the cells from red to far-red light, we observed a decrease in GP as soon as 5 min, which was the imaging frequency. Likewise, the GP increased already after 10 min changing the illumination from far-red to red light. We added the time course measurements to the manuscript (**Supplementary Fig. 5a**). In agreement with the GP measurements, AFM measurements taken after 10 min red or far-red light illumination gave consistent results in the measured forces. See changes in the text above.*

Comment:

On line 153 they state that “Cph1-PM mediated changes in membrane tension affect membrane to cortex attachment and formation of focal adhesions”. This suggests that increase in tension happens before changes in the cortex and the formation of focal adhesions. Is it possible to address the dynamics of these changes? I understand it may be difficult to address the timecourse of events. If so, the authors need to soften some of their statements which suggest cause and effect.

Response:

Indeed, it is difficult to entangle the exact sequence of events after the formation of the artificial cell-cell adhesions. Following the recommendation, we soften the statement as follows:

“Cph1-PM mediates changes in membrane tension, membrane-to-cortex attachment and formation of focal adhesions”.

Additionally, we added the following point to the discussion:

“However, although this study identifies constituents that mediate this collective cell motion, the precise sequence of signal propagation through the system remains elusive. Unlike the switch-like activation of Cph1-PM, which occurs at a timescale of milliseconds³⁴, the formation of cell-cell

adhesions depends on the time that two cells require to come into contact. The resulting changes in membrane tension, which occur at timescales of minutes but may last for hours, will then affect systems that operate at faster timescales (e.g., steady-state turnover of the cell cortex and lipid metabolism), as well as those that operate at slower timescales (e.g., gene transcription and protein synthesis). Consequently, future work will be needed to delineate the exact cause and effect of the observed changes reported in this study.”

Comment:

The authors observe an increase in ERM phosphorylation after illumination. Do they still observe an increase in tension when removing the membrane cortex crosslinking? How much of the tension increase is related to the increased activity of Cph1 and how much of it is related to downstream signaling?

Response:

It is undoubtedly intriguing to determine how much of the increase in membrane tension is directly attributable to Cph1-based adhesions versus downstream signaling. However, this poses a significant technical challenge, if not an impossibility, as there are no clear methods to selectively disrupt membrane cortex crosslinks without simultaneously altering the actin cytoskeleton or membrane structure.

Comment:

The authors analysed changes in E-cadherin after illumination. However, what about other cadherins, such as N-cadherin? Presumably, if these cells are mesenchymal then they should still have N-cad. And just because they do not observe a change in levels of cadherins, it doesn't mean that adhesion proteins may not also be involved in the response.

Response:

MDA-MB-231 cells, unlike some other breast cancer cell lines, have been shown to lack type 1 cadherins, including E-cadherin and N-cadherin. We now highlight this point in the text. Our experiments further demonstrate that Cph1-PM-MDA cells exhibit weak native adhesions, comparable to those of MDA-MB-231 cells under far-red light illumination. The data indicate that these adhesions do not lead to increased E-cadherin expression, which would otherwise promote collective migration. Additionally, the newly included wound healing assays, where cells are switched from red to far-red light after 6 hours, provide further evidence that cell-cell adhesions are predominantly mediated by Cph1.

“As a starting point, we used MDA-MB-231 (MDA) cells, which lack type-1 cadherins, including E- and N-cadherin; display weak cell-cell adhesions; and migrate as single cells²⁸.”

Comment:

They chemically perturb the actin network and examine wound closure. However, does perturbing actin also effect membrane tension after red light illumination?

Response:

We find that perturbing actin independent of the light illumination inhibits wound closure (Fig. 3b, d). Therefore, we also expect that perturbing actin itself would alter membrane tension and the insight gained with red light illumination would not provide further insight.

Comment:

There is no reference to movie 4. And in line 265 they incorrectly reference video 4 when they should instead reference video 5.

Response:

We corrected this point. We now refer to all movies in the text in the correct order.

Comment:

In movie 4 and 5 the cell densities look different in the far red vs red treatments. The red cell densities are higher. Can we be sure that the differences in migration are not related to differences in starting cell densities?

Response:

In the previously analyzed movie, some dead cells resulting from the wounding process adhered to the top of the cell layer. To minimize the impact of cell density, we seeded the same number of cells in all experiments. The Supplementary Movies have been replaced with versions that exhibit similar cell densities.

Comment:

They directly affected membrane tension through the addition of oleic acid (OA), which led to an increase in migration speed without an increase in cell correlation. Doesn't this suggest that the increase in speed can be uncoupled from the increase in cell coordination in response to an increase in membrane tension? The following analysis of PI3K/mTor appears to suggest that increased speed and cell coordination are in response to a single pathway downstream of the increase in membrane tension. But maybe membrane tension has pleiotropic effects on the cell.

Response:

The reviewer is right in that membrane tension alone increases migration speed through the downstream PI3K/mTor pathway. Indeed, a previous study demonstrated that oleic acid (OA) increases the wound healing in breast cancer cells, particularly MDA-MB-231, and that this increase operated through the upregulation of phospholipase D2 (PLD2) and subsequently increasing mTOR activity. It is correct that the artificial cell-cell adhesions change the cell migration both in terms of collectivity and speed. We discuss this point in the manuscript:

"Previous studies have demonstrated that OA enhances the migration speed of individual MDA cells by elevating PLD2 levels and activating downstream mTOR signaling⁴⁹. Consistent with these findings, our results show that artificial cell-cell adhesion influences cell migration by promoting both collective behavior and individual cell speed, driven by an increase in membrane tension."

Comment:

Inhibition of PTEN with SF1670 resulted in an increased coordination of the cells under control conditions. They also state that SF1670 increased the rate of wound closure in control cells. Is this counter to their initial argument that increased collective interactions should slow down migration? Any idea how PTEN inhibition is doing this?

Response:

We thank the reviewer for pointing out this point. We found a report stating that "... loss of tumour suppressor PTEN alone is sufficient to enhance the collective migration of glial cells in vitro and EC in vivo. This effect is independent of PI3K/AKT signalling, but requires LKB1-dependent activation of AMPK, a master regulator of metabolism." (Peglion F, et al. PTEN inhibits AMPK to control collective migration. Nature Communications 13, 4528 (2022).)

Given this influence of PTEN, that is independent of the PI3K/AKT signaling, we decided to remove this data and related discussion to avoid confusion and giving a false impression.

Comment:

I am not sure I understand the interpretation of mTor inhibition with rapamycin. They state that the rate of wound closure of Cph1/red cells decreased after rapamycin, arguing that inhibition of the mTor pathway selectively suppressed the collective behavior of the cells. What is the relationship of wound closure speed and collective motion?? And in figure 5 they do not quantify cell collectivity, only speed.

Response:

We agree with the reviewer that the data shows that cell migration speed and not the collectivity is altered through the modulation of the mTOR pathway. We changed our statement accordingly. "These data suggest that the observed increase in cell motility of the Cph1-PM-MDA cells, induced by the artificial cell-cell adhesions under red light, is mediated through the mTOR signaling pathway."

Figure comments

Comment(s):

- Fig 1b (and 3d, 3h, 3i, 4a, 4e, 4i, 5b): please display as box and whiskers plus data points or datapoints only (with error bars) - not bar plot.
- Fig 1c (and 3e, 4f, 4j): display single data points overlaid to box and whiskers plot

Response:

Following the journal guidelines, we show individual data points for data sets with up to 10 data points. We prefer to use bar plots for data sets with up to 10 points as box plots with medians are not meaningful for small data sets. For data sets with 10-100 data points, we use box plots including single data points. For data sets with more than 100 data points, we use box plots without showing the individual data points, as the points are not properly resolved and make the figures look cluttered. We provide all data points in the source data.

Comment:

Fig 1d (and 3b, 4c, 4d, 4h): what is the PIV measured on? Time 3h vs time 0? Are velocity vectors normalised and how? What min/max represent in the colour bar? No details in caption, please fix.

Response:

Particle Image Velocimetry (PIV) plugin measures the movement of fluids or particles by calculating vector fields from images. In our experiments, we applied PIV to track cell movement over time by analyzing images taken at regular intervals (10 minutes). This plugin calculated the velocity vectors by determining the change in position of particles across successive images, considering both the distance traveled and the time elapsed. The velocity vector data included both the direction and speed of movement.

*In the generated vector field, the minimum and maximum values represent the range of cell movement speeds from images. The minimum value corresponds to the smallest movement, while the maximum value represents the area where the cells exhibit the fastest movement. Our analysis revealed that regions with the maximum values, indicating the fastest cell movement, were more prevalent under red light, whereas the far-red light exposure showed a higher concentration of minimum values, indicating slower movement. The color vector generation codes are provided in **Supplementary Code 2**. The bright field images at 0 hours and 3 hours are presented to demonstrate how the cells migrated over time. We added the following sentences in "Methods" section:*

*“In our experiments, we applied PIV to track cell movement over time at regular intervals (10 minutes). Custom code was used to create correlation lengths (**Supplementary Code 1**), colored vector plots (**Supplementary Code 2**), and rose vector plots (**Supplementary Code 3**)⁹.”*

Comment(s):

- Fig 1e: what is a directional rate vector? Are these calculated from the PIV or separately? Is this a single particle tracking as suggested by the colour bar legend? No details in the methods.
- Fig 1f: what is the value assigned to one cell diameter? What is a ‘straightness index’ and how is it calculated? Again, no details in the methods, please fix.
- Fig 1g: What is a ‘straightness index’ and how is it calculated? Again, no details in the methods, please fix. Please display single data points overlaid to box and whiskers plot.

Response:

*In **Fig. 1e**, we performed the assay using a live cell compatible nuclear stain. Thereby, we were able to track the movement of individual cells over time. This is an independent method to evaluate coordinated cell movement from the PIV in **Fig. 1d**. We added the details of the single cell tracking and subsequent analysis to the methods.*

“To analyze the migration pattern during wound closure, single-cell tracking was performed using the TrackMate Plugin (v7.11.1)⁵⁸ in ImageJ (1.54f). For particle detection and linking, a Laplacian of Gaussian (LoG) detector and a Simple Linear Assignment Problem Tracker were used, respectively. Detection of the wound edge and the analysis of migration parameters including cells speed, straightness index and the orientation of migration were performed in MATLAB (R2020b).

To distinguish between cells located close to the wound edge and cells further back in the field, a line parallel to the wound edge was drawn with a distance of 60 μm , which is approximately the size of three nuclei. If the center of a nuclei was located within this region, a cell was considered to be migrating at the front. Trajectories were cut off before cells from opposite sides of the wound were touching each other, trajectories shorter than four steps were excluded from analysis.

The straightness index was determined as the ratio between the net displacement and the total distance traveled, ranging from 0 (= movement back to the origin) to 1 (=straight trajectory). The cell speed represents the median speed of each cell in the respective time interval. As an indicator of directionality, the angle of migration in relation to the wound edge is used, referred to as the orientation angle. Angles around zero degree indicate migration towards the open space, the wound edge is here considered to be a straight vertical line.”

Comment:

Fig 1i: display single data points overlaid to box and whiskers plot. Figure panel not mentioned in the text.

Response:

*We show the distribution in the cell speed values including their spatial distribution in **Fig. 1i** as a pseudo color image in **Fig. 1h**. As explained above, we don’t include the individual data points in the box plots, as there are about 2000 data points. We now mention **Fig. 1i** in the text, and thank the reviewer for pointing it out.*

Comment:

Fig 2a,b: how is the normalized general polarization calculated/measured?

Response:

*General polarization (GP) measures the emission spectral shift, for which the formula is provided in the methods. To calculate the GP values, we developed a code in ImageJ, which we provide now as **Supplementary Code 4**.*

Comment:

Fig 2c: In the methods they state “contact time was varied between 100 ms to 10 s, aiming at maximizing the probability of extruding single tethers” – this is a big time range, can they please elaborate further? Is this standard for this assay?

Response:

During atomic force microscopy experiments, the cantilever was coated with Concanavalin A to facilitate nonspecific binding to glycoproteins on the cell membrane. The formation of tethers from cantilever attachment to the membrane is highly dependent on the contact time, which is critical for ensuring proper interaction. Initially, Concanavalin A provides effective binding, and a contact time of approximately 100 ms is typically sufficient to form tethers. However, over the course of multiple measurements, the effectiveness of Concanavalin A may decrease, making it difficult to form tethers within such a short, millisecond-scale time frame. In these cases, increasing the contact time — up to 10 seconds — can improve the cantilever’s attachment to the cell membrane. Consequently, the optimal contact time will vary depending on experimental conditions and should be adjusted accordingly for each experiment. Further details on the contact time were provided in the previous study.

Bergert M, Diz-Muñoz A. Quantification of Apparent Membrane Tension and Membrane-to-Cortex Attachment in Animal Cells Using Atomic Force Microscopy-Based Force Spectroscopy. In: Mechanobiology: Methods and Protocols (ed Zaidel-Bar R). Springer US (2023).

Comment:

Fig 2g (and 3g, 4a): how was this quantified?

Response:

*The image analysis for pERM in **Fig. 2g** was similar to that of the FA in **Fig. 2j**, I and is detailed in the response below.*

*For **Fig. 3g**, we added a more detailed description of the image analysis pipeline to the manuscript: “For YAP analysis, a mask was created using DAPI staining. The intensity of the YAP inside and outside the nuclei was measured using the DAPI mask, and the nuclear signal was divided by the cytoplasmic signal.”*

*For **Fig. 4a**, we followed the manufacturer’s protocols. We added the protocol to the Methods section:*

“Total phosphatidic acid was measured with an assay kit purchased from Cell Biolabs, Inc. and run according to the manufacturer’s instructions. In brief, 5×10^6 cells were plated in triplicate in the presence of PCB and either with or without PLD2 modulating drugs, and cultured overnight. The cells were illuminated for 4 hours before the cells were washed three times with ice-cold PBS for 5 minutes each time. Then, cells were collected via scraping over ice. Cells were pelleted and resuspended in 1 mL of PBS before being sonicated for 30 sec at 1 sec pulses at 30% power. For lipid extraction, 1.5 mL methanol, 2.25 mL 1 M NaCl, and 2.5 mL chloroform were added, vortexed, and centrifuged at $1500 \times g$ for 10 minutes at 4°C. The upper aqueous phase was discarded, and the lower chloroform phase was washed twice with pre-equilibrated upper phase, then transferred to a vial. The solvent was removed from the samples and then resuspended in 50 μ L an assay buffer. 10 μ L of each sample was transferred to a 96 well plate and incubated with 40 μ L of Lipase Solution at 37 °C for 30 min. 50 μ L of the Detection Enzyme Mixture provided with the kit was added and the samples were incubated at room temperature for 10 min in the dark.

The concentration of PA was determined by measuring the fluorescence intensity using excitation at 545±15nm and emission at 590±5nm. The PA contents were calculated based on PA standards provided by manufacturer.”

Comment:

Fig 2i: how was cell area quantified? Was actin used as a proxy?

Response:

Yes, actin was used to define the boundaries of cells, as described in the methods.

Comment:

Fig 2j-l: how was FA area quantified?

Response:

A detailed description of the image analysis for vinculin and pERM stained cells was added to the revised manuscript.

“To measure the area of vinculin and pERM staining in cells, the cell boundaries were defined using phalloidin staining. The vinculin and pERM signals were analyzed in each cell according to established protocols⁶². To improve image analysis, the background was first removed, and local contrast was enhanced using CLAHE (Contrast Limited Adaptive Histogram Equalization). An exponential function (exp) was then applied to further minimize background interference. Brightness and contrast were automatically adjusted to optimize feature visibility. Finally, the grayscale image was converted into a binary format, and objects with an area greater than 0.5 μm^2 were identified and counted for each cell.”

Comment:

Fig2: check panel letters in legend

Response:

All panel letters of the figures were double-checked.

Comment:

Image analysis methods lack details. Please describe the analysis better and provide strategies and chosen parameters for all the software tools used, keeping in mind a reader should be able to replicate your results. For example, “For wound healing, images were analyzed with ImageJ (National Institute of Health, USA) to determine the area of the wound at each time point”: how was the analysis performed? Which plugin tools were used if any?

Response:

We added a detailed explanation of the image analysis to the methods section (see Comment above).

Minor comment:

Can you please make sure that the order of figure panels follows the text and the alphabetical order?

Response:

Following the suggestion, we adjusted the numbering and labels of all figures.